# Effective Human-AI Teams via Learned Natural Language Rules and Onboarding

**Hussein Mozannar**[1,2]    **Jimin J Lee**[1,2]    **Dennis Wei**[1,3]
**Prasanna Sattigeri**[1,3]    **Subhro Das**[1,3]    **David Sontag**[1,2]
[1]MIT-IBM Watson AI Lab, Cambridge, MA
[2]CSAIL and IMES, Massachusetts Institute of Technology, Cambridge, MA
[3]IBM Research, Cambridge, MA
{mozannar;gminnout}@mit.edu, {dwei;psattig}@us.ibm
subhro.das@ibm.com, dsontag@mit.edu

## Abstract

People are relying on AI agents to assist them with various tasks. The human must know when to rely on the agent, collaborate with the agent, or ignore its suggestions. In this work, we propose to learn rules grounded in data regions and described in natural language that illustrate how the human should collaborate with the AI. Our novel region discovery algorithm finds local regions in the data as neighborhoods in an embedding space that corrects the human prior. Each region is then described using an iterative and contrastive procedure where a large language model describes the region. We then teach these rules to the human via an onboarding stage. Through user studies on object detection and question-answering tasks, we show that our method can lead to more accurate human-AI teams. We also evaluate our region discovery and description algorithms separately.

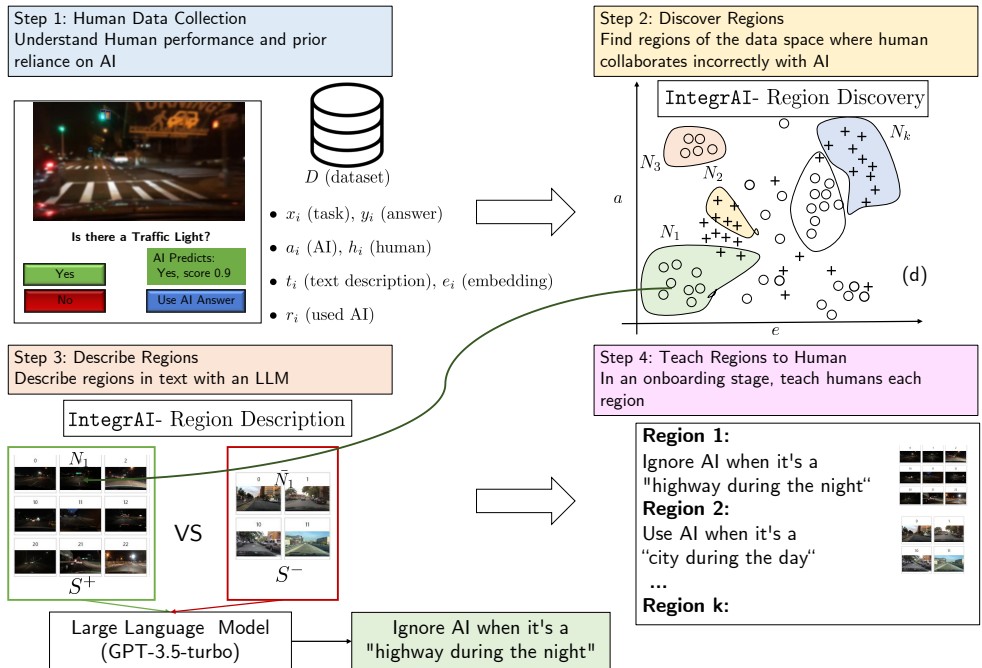

Figure 1: The proposed onboarding approach with the `IntegrAI` algorithm.

37th Conference on Neural Information Processing Systems (NeurIPS 2023).

# 1 Introduction

As artificial intelligence (AI) becomes more ubiquitous and sophisticated, humans are increasingly working alongside AI systems to accomplish various tasks, ranging from medical diagnosis [9, 28], content moderation [30] to writing [21] and programming [22]. One of the promises of AI is to enhance human performance and efficiency by providing fast and accurate solutions. However, the literature on human-AI collaboration has revealed that humans often underperform expectations when working with AI systems [36, 49, 40, 77]. Moreover, studies have shown that providing explanations for the AI's predictions often does not yield additional performance gains and, in fact, can make things worse [49, 15, 32]. These negative results of human-AI performance may be attributed to two possible reasons. First, humans have miscalibrated expectations about AI's ability, which can lead to over or under-reliance [5]. Second, the cost of verifying the AI's answer with explanations is often higher than that of solving the task without the AI as some explanations don't help in verifying the AI answer [75].

The central question remains: how can we collaborate better with AI models? In this work, we propose an intuitive framework for thinking about human-AI collaboration where the human first decides on each example whether they should rely on the AI, ignore the AI, or collaborate with the AI to solve the task. We refer to these three actions as the *AI-integration decisions*. The human-AI team will perform optimally if the human knows which action is best on a task-by-task basis. We propose `IntegrAI` (Figure 1), an algorithm that leverages data from baseline human interactions with the AI to learn near-optimal integration decisions, in the form of *natural language rules* that are easily understandable. These rules are then taught to the human through an onboarding stage, analogous to an onboarding class that humans might take before operating machines and equipment. Onboarding additionally calibrates the human's expectations about AI performance. We further investigate surfacing the AI-integration decisions found by `IntegrAI` as recommendations to the human within an AI dashboard used after onboarding. The hope is that onboarding and the dashboard help the human know which action they should take, thereby leading to effective AI adoption for enhanced decision-making.

Learning AI-integration rules requires a dataset of paired examples and human predictions (Figure 1 Step 1). Each rule is defined as a bounded local region centered around a learned point in a potentially multi-modal embedding space spanning the task space and natural language (Figure 1 Step 2). For example, CLIP embeddings [62] connect image and text spaces for tasks involving images, and typical text embeddings [64] are used for natural language tasks such as question answering. The regions are obtained with a novel region discovery algorithm. Then, a text description of the region is generated, resulting in a rule that indicates whether the human should ignore, rely on, or collaborate with the AI. We obtain descriptions using a novel procedure that connects the summarization ability of a large language model (LLM) [14] with the retrieval ability of embedding search to find similar and dissimilar examples. The procedure first queries the LLM to describe points inside the region (Figure 1 Step 3). The embedding space is then leveraged to find counterexamples inside and outside the region to refine the description.

We first evaluate the ability of our region finding and region description algorithms to find regions that will aid the human-AI team in several real-world datasets with image and text modalities. We then investigate the efficacy of both algorithms in synthetic scenarios where we know the ground truth regions. Finally, we conduct user studies on tasks with real-world AI models to evaluate our onboarding and AI-integration recommendation methodology. Our main task is detecting traffic lights in noisy images [83] from the perspective of a road car, motivated by applications to self-driving cars. The user study reveals that our methods significantly improve the accuracy of the human-AI team by 5.2% compared to without onboarding. We investigate a second task of multiple choice question answering using the MMLU dataset [33] and find that onboarding has no effect on performance and that only displaying AI-integration recommendations has a negative effect. To summarize, the key contributions of this paper are as follows:

- We propose a novel region-finding and region-description algorithm, `IntegrAI`, that outperforms prior work to derive rules that can guide humans in collaborating with AI models for shared decision-making.
- We demonstrate the effect of onboarding and displaying AI-integration recommendations on two real-world tasks, and find that onboarding has a significant positive effect on one task and that integration recommendations are not useful.

## 2 Related Work

A growing literature of empirical studies on AI-assisted decision making has revealed that human-AI teams do not perform better than the maximum performance of the human or AI alone even with AI explanations [7, 71, 49]. Recent work has explored modifying AI explanations to depend on human behavior [77, 51] and showed positive results with synthetic AI models. A growing literature exists on onboarding humans to work on AI models [67, 17, 45, 57, 16, 41], however, our work enables the automated creation of onboarding without any human in the loop. We compare to a representative from work on discovering regions of errors in AI models [24, 10, 66, 82, 87, 80, 23, 63, 37], our work instead focuses on regions of disparate performance between human and AI. Learning to defer methods learn models that decide using a secondary AI model who between the human and the AI classifier should predict [52, 56], this paper focuses on the reverse setting where the human makes all decisions but utilizes some of the thinking from that literature. Our AI-integration recommendations are also related to personalized policies [11]. Our MMLU experiments share similarities with recent work [13, 86, 74, 38]. Further comparison to prior work can be found in Appendix A.

## 3 AI Assisted Decision Making

**Setting.** We consider a setting where a human is making decisions with the help of an AI agent who provides advice to complete a **task**. Formally, the **human** has to make a decision $Y \in \mathcal{Y}$ given access to information about the context as $Z \in \mathcal{Z}$ and the AI's advice $A \in \mathcal{A}$. We denote the human as a potentially randomized function $H(Z, A; \theta_h)$ with parameters $\theta_h$ which are unobservable. On the other hand, the **AI** agent provides advice based on its viewpoint of the context as $X \in \mathcal{X}$, which might be different from the humans viewpoint $Z$ as the human may have side information. The AI model provides advice according to $M(X; \theta_m) := A \in \mathcal{A}$, the advice always includes a candidate decision $\hat{Y}$ and possibly an explanation of the decision. We assume that the observed tasks are drawn from an underlying distribution, $\mathbb{P}_{X,Z,Y}$, over the contexts of AI and human, and the ground truth. For simplicity of exposition, we will assume that $X = Z$ when describing our methods.

**Task Metrics.** The human wants to make a decision that optimizes various metrics of interest. Given a ground truth decision $Y$ and a chosen decision $\hat{Y}$, the loss is given by $l(Y, \hat{Y}) : \mathcal{Y} \times \mathcal{Y} \to \mathbb{R}^+$. We denote the loss $L(H, M)$ of the Human-AI team over the entire domain as:

$$L(H, M) := \mathbb{E}_{x,z,y \sim \mathbb{P}} \left[ l \left( (y, H \left( z, M(x; \theta_m); \theta_h \right) \right) \right] \tag{1}$$

In our example, this could be the 0-1 loss $\mathbb{I}_{Y=\hat{y}}$. We are further interested in metrics that convey efforts undertaken by the human during the process. Particularly, we focus on *time to make a decision*, which can be measured when the human makes decisions.

**Human-AI team.** The decision of the Human-AI team is represented by the function $H(Z, A; \theta_h)$. The human without the AI is denoted by $H(Z, \emptyset; \theta_h) := H(Z; \theta_h)$, which is obtained by setting the AI advice to the empty set, i.e. no advice. In theory, we expect the human with advice to perform at least as well as without advice, simply because the human can always ignore the advice. However, the literature on Human-AI teams has clearly demonstrated that this is often not the case [46]. An **effective** Human-AI team is one where the human with the AI's advice achieves a more favorable trade-off in terms of the metrics than without the advice.

**Framework for Cooperation.** Our insight into forming an effective team is to explicitly recommend when the human should consider the advice and how to incorporate it into their decision. We propose a two-stage framework for the human to cooperate with the AI: the human first decides whether to ignore the AI advice, use the AI's decision or integrate the AI advice to make a decision with explicit cooperation. Each of these three cases provides a clear path to the second stage of making the final output decision.

**Definition 1.** *The AI-integration function $R(Z, A; \theta_r)$, also referred to as integrator, formalizes a framework for the human to cooperate with the AI:*

$$R(Z, A; \theta_r) = \begin{cases} 0 & \to H(Z; \theta_h) & \textit{(ignore AI)} \\ 1 & \to \hat{Y} & \textit{(use AI decision as is)} \\ 2 & \to H(Z, A; \theta_h) & \textit{(collaborate with AI)} \end{cases} \tag{2}$$

*In this work, we only consider the actions of ignoring or using the AI decision: $R \in \{0, 1\}$ and leave the action of $R = 2$ for future work.*

The integration function can be thought of as a specific formalization of the human mental model of the AI [5, 6]. Given an integration function $R$ and a human $H$, we can define a hypothetical human decision maker $H_R$ who first computes $R$ and then follows its recommendation to make a final decision. Similarly, for each human $H(Z, A; \theta_h)$, we can associate an integration function $R_H$, such that $H_{R_H} = H$. By fixing $H(Z; \theta_h)$, one can try to minimize the loss $L(H_R, M)$ over all possible choices of $R$, an optimal point of such an integration function is denoted $R^*$. Following the recommendations of the optimal AI-integration function leads to an **effective** Human-AI team.

**Learning Rules and Onboarding.** There are two problems that need to be solved to achieve this vision: how do we learn such an $R^*$ and how can we ensure that the human follows the recommendations of this $R^*$? In the next section, we outline how we approximate the optimal integration function; this is fundamentally a machine-learning problem with its own challenges that we tackle in Section 4. The second obstacle is that the human should know to follow the recommendations of $R^*$. To ensure this, we propose an **onboarding** stage where the human learns about the AI and the optimal integration function and additionally display the recommendations as part of an AI dashboard. In this onboarding stage, we will help the human shape their internal parameters $(\theta_h, \theta_r)$ to improve performance. This is a human-computer interaction (HCI) problem that we tackle in Section 5.

## 4    Learning Rules for Human-AI Cooperation: `IntegrAI`

In this section, we discuss how to learn an *integrator* function $\hat{R} : \mathcal{X} \times \mathcal{A} \to \{0, 1\}$ to approximate an optimal integrator while being understandable to the human. We first describe the ingredients for this learning (integrator as a set of regions, objective function, dataset) before detailing how we learn regions and describe them in Sections 4.1 and 4.2 respectively.

**Integrator as a Set of Regions.** Since the integrator $\hat{R}$ will be used to both onboard the human and provide test-time recommendations as part of an AI dashboard, it should be easily understandable to humans. If the goal was to build the most accurate integrator, we could use work on learning to defer [56, 19, 58]. To address this requirement, we propose to parameterize the integrator in terms of a set of local data regions, each with its own integration decision label as well as a natural language description. More specifically, we aim to learn a variable number of regions $N_1, N_2, \cdots$ as functions of $(X, A)$, the observable context and the AI's advice. Each region $N_k$ consists of the following: 1) an indicator function $I_{N_k} : \mathcal{X} \times \mathcal{A} \to \{0, 1\}$ that indicates membership in the region, 2) a textual description of the region $T_k$, and 3) a takeaway for the region consisting of an integration decision in $r(N_k) \in \{0, 1\}$. We additionally want these regions to satisfy a set of constraints so that they are informative for the human and suitable for onboarding.

**Maximizing Human's Performance Gain.** Since we are working with human decision-makers, we have to account for the fact that the human implicitly has a **prior** integration function $R_0$, which represents how the human would act without onboarding. Thus, in learning integrator regions, our goal is to maximize the human performance gain relative to their prior. The performance gain is defined as follows for points in a region $N$:

$$G(N, \hat{R}, R_0) = \sum_{i \in N} l(y_i, H_{R_0}(x_i, a_i)) - l(y_i, H_{\hat{R}}(x_i, a_i)), \tag{3}$$

where $l$ is the loss defined in Section 3. Note that the notion of a human's prior mental model was also discussed by [57] but we expand on the notion and are able to learn priors as we discuss below.

**Dataset with Human Decisions.** We assume we have access to a dataset $D = \{x_i, y_i, h_i, r_{0i}\}_{i=1}^{n}$ sampled from $\mathbb{P}$, where $x_i$ is the AI-observable context, $y_i$ is the optimal decision on example $i$, $h_i$ is a human-only decision, defined in Section 3 as $H(x_i; \theta_h)$, and $r_{0i} \in \{0, 1\}$ is an indicator of whether the human relied on AI on example $i$. We thus regard the samples $\{r_{0i}\}_{i=1}^{n}$ as a proxy for prior human integration function $R_0$. The prior integration decisions of the human $r_{0i}$ are collected through a data collection study where the human predicts with the AI without onboarding. When the human presses on the "Use AI" button (see Figure 3) we record $r_{0i} = 1$, and 0 otherwise. The human predictions $h_i$ are collected through a secondary data collection study where the human makes predictions without the AI. We also assume that we are given an AI model $M$, from which we obtain

AI decisions $\hat{y}_i \in a_i$ from the AI advice $a_i = M(x_i; \theta_m)$ [1]. Given the dataset $D$, AI decisions $\hat{y}_i$, and loss $l(.,.)$, we can define optimal per-example integration decisions $r_i^*$ by comparing human and AI losses on the example: $r_i^* = \mathbb{I}\left(l(y_i, h_i) > l(y_i, \hat{y}_i)\right)$.

## 4.1 Region Discovery Algorithm

**Representation Space.** In this subsection, we describe a sequential algorithm that starts with the prior integration function $R_0$ and adds regions $N_k$ one at a time. The domain $\mathcal{X}$ for the task may consist of images, natural language, or other modalities that possess an interpretable representation space. We follow a similar procedure for all domains. The first step is to map the domain onto a potentially cross-modal embedding space using a mapper $E(.)$, where one of the modes is natural language. The motivation is that such an embedding space will have local regions that share similar natural language descriptions, enabling us to learn understandable rules. For example, for natural images, we use embeddings from the CLIP model [62]. The result of this step is to transform the dataset $\{x_i\}_{i=1}^n$ into a dataset of embeddings $\{e_i\}_{i=1}^n$ where $e_i \in \mathcal{E}_\mathcal{X} \subseteq \mathbb{R}^d$.

**Region Parameterization.** We define region $N_k$ in terms of a centroid point $c_k$, a scaled Euclidean neighborhood around it, and an integration label $r_k \in \{0, 1\}$. The neighborhood is in turn defined by a radius $\gamma_k \in \mathbb{R}$ and element-wise scaling vector $w_k$. Both $c_k$ and $w_k$ are in $\mathcal{E}_\mathcal{X} \times \mathcal{A}$, the concatenation of the embedding space and the AI advice space. The indicator of belonging to the region is then $I_{N_k}(e_i, a_i) = \mathbb{I}_{||w_k \circ ((e_i, a_i) - c_k)||_2 < \gamma_k}$, where $\circ$ is the Hadamard (element-wise) product.

**Region Constraints.** We add the following constraints on each region $N_k$ to make them useful to the human during onboarding. First, the region size in terms of fraction of points contained must be bounded from below by $\beta_l$ and above by $\beta_u$. Second, the examples in each region must have high agreement in terms of their optimal per-example integration decisions $r_i^*$. Specifically, at least a fraction $\alpha$ of the points in a region must have the same value of $r_i^*$.

`IntegrAI`**-Discover.** In round $k$, we add a $k$th region to the integrator $R_{k-1}$ from the previous round (with $k - 1$ regions) to yield $R_k$. After $m$ rounds, the updated integrator $R_m$ is defined as follows: Given a point $(x, a)$, if it does not belong to any of the regions $N_1, \ldots, N_T$, then we fall back on the prior. Otherwise, we take a majority vote over all regions to which $(x, a)$ belongs:

$$R_T(x, a) = \begin{cases} R_0(x, a) & \text{if } I_{N_k}(x, a) = 0, \ k = 1, \ldots, T \\ \text{majority}\left(\{r(N_k) : k \text{ s.t. } I_{N_k}(x, a) = 1\}\right) & \text{otherwise.} \end{cases} \quad (4)$$

In round $k$, we compute the potential performance gain for each point if we were to take decision $r$ on the point as $g_{i,r} = l(y_i, H_{R_{k-1}}(z_i, a_i)) - l(y_i, r)$. The optimization problem to find the optimal regions is non-differentiable due to the discontinuous nature of the region indicators. To make this optimization problem differentiable, we relax the constraints as penalties with a multiplier $\lambda$ and replace the indicators with sigmoids scaled by a large constant $C_1$:

$$\max_{c, \gamma, w, r} \ \sum_{i=1}^n \sigma(C_1(-||w \circ ((e_i, a_i) - c)|| + \gamma)) \cdot g_{i,r} - \lambda \max(\sum_{i=1}^n \sigma(C_1(-||w \circ ((e_i, a_i) - c)|| + \gamma))$$

$$\cdot (\mathbb{I}_{r_i^* = r} - \alpha n), 0) - \lambda \max\left(\sum_{i=1}^n \sigma(C_1(-||w \circ ((e_i, a_i) - c)|| + \gamma)) - \beta_u n, 0\right)$$

The lower bound constraint with $\beta_l$ can be added in a similar fashion. We find the value of $r$ with an exhaustive search and use a gradient based procedure for the other optimization variables. The full optimization details are in Appendix B.

## 4.2 Region Description Algorithm

We now describe our region description algorithm aimed at making the rules for integration human understandable. Natural language descriptions are a good match for this objective. Specifically, we would like to find a contrastive textual description $T_k$ of each region $N_k$ that describes it in a way to distinguish it from the rest of the data space.

**Textual Descriptions for Regions:** The first step is to have a textual description $t_i$ for each example in our dataset $D$ based on $x_i$. If textual descriptions are not available, we can obtain them by utilizing

---

[1]Moreover, we assume the AI was not trained on the dataset $D$ so that we can use $D$ to obtain an unbiased measurement of AI performance; this is crucial, as otherwise, we might overestimate its performance.

---

**Algorithm 1** `IntegrAI`-Describe

---

**Input**: Dataset $D$, region $N_k$

1: $S^+ \leftarrow 15$ random examples from $N_k$, $S^- \leftarrow 5$ random examples outside $N_k$
2: **Initial Region Description:** $T_k^0 \leftarrow \hat{O}(S^+, S^-)$ (LLM call)
3: **for** $i = 1, \cdots, m$ **do**
4:     **Find Counterexample outside region.** $s^- = \arg\max_{j \notin N_k} \text{sim}(E(T_k^i), e_j)$ (most similar outside region)
5:     **Find Counterexample inside region.** $s^+ = \arg\min_{j \in N_k} \text{sim}(E(T_k^i), e_j)$ (least similar inside region)
6:     **Update Inside and Outside Sets.** $S^- \leftarrow S^- \cup t_{s^-}$ and $S^+ \leftarrow S^- \cup t_{s^+}$
7:     **Get New Region Description:** $T_k^i \leftarrow \hat{O}(S^+, S^-)$ (LLM call)
8: **end for**

**Return:** Region description $T_k^m$

---

models that map from the domain $\mathcal{X}$ to natural language such as captioning models for images, summarization models for text, or exploiting metadata to construct a natural language description. One idea to obtain a region description is to ask an LLM (such as GPT-3.5) to summarize all textual descriptions of points inside the region. However, there are two issues with this approach: first, the region may contain thousands of examples so we need an effective way to select which points to include, and second, the obtained region description may not contrast with points outside the region. To resolve these issues, we propose an algorithm that iteratively refines region descriptions with repeated calls to an LLM ($\hat{O}$) in 1 (`IntegrAI`-Describe). The algorithm starts with an initial description and then finds counterexamples to that description at each round: examples outside the region with high cosine similarity with respect to embeddings (sim) of the region description and examples inside the region with low similarity to the region description. Then we add those counterexamples to our example sets and derive a new region description, we use an especially created prompt with an exemplar to the LLM to get the region description at each round, the prompt can be found in Appendix C.

**Illustrative Example.** Suppose we want to describe a region consisting of images of highways during the night, with no cars present (see Figure 1 for images of BDD [83]). Our method's initial description is "The highway during the night with clear, rainy or snowy weather", not mentioning that the highway has no cars, particularly because the captions of examples $t_i$ only mention the presence of cars and not their absence. In the second round, the algorithm finds the counterexample $s^-$ as 'city street during the night with clear weather with a lot cars' and counterexample $s^+$ "highway during the night with clear weather". The new description $T_k^1$ becomes "clear highway during the night with various weather conditions, while outside the region are busy city street at night with clear weather". After one more round, the description $T_k^2$ becomes "driving on a clear and uncongested highway during the night in various weather conditions". We now proceed in the next section to describe how we onboard the human decision maker using the regions.

## 5 Onboarding and Recommendations to Promote Rules

Once rules for integration have been learned as described in the previous section, the task is to teach these rules to the human with an **onboarding** stage and encourage their use at test time. We accomplish this through an **onboarding** process followed by test-time **recommendations**, as described next.

**Human-AI Card.** The onboarding process accordingly consists of an *introductory* phase and a *teaching* phase. In the *introductory phase*, the user is first asked to complete a few practice examples of the task on their own to gain familiarity. The user is then presented with general information about the AI model in the form of a **human-AI card**, akin to a model card [55], that includes AI model inputs and outputs, training data, training objectives, overall human and AI performance along with AI performance on subgroups that have performance that deviate from average performance (details in Appendix D).

The **teaching phase** is structured as a sequence of "lessons", each corresponding to a region $N_k$ resulting from the algorithm of Section 4.1. The specific steps in each lesson are as follows:

**Step 1: Human predicts on example.** A representative from the region is selected at random that has an optimal integration decision identical to that of the region. The human is asked to perform the task for the chosen representative and is shown the AI's output along with the option to use the AI's response.

**Step 2: Human receives feedback.** After submitting a response, the user is told whether the response is correct and whether the AI is correct.

**Step 3: From Example to Region Learning.** The user is informed that the representative belongs to a larger region $N_k$ and is provided with the associated recommendation $r_k$, textual description $t_k$, and AI and human performance in the region as well as the raw examples from the region in a gallery viewer.

After completing all lessons as above, a second pass is done where the user is re-shown all lessons for which their response was incorrect. This serves to reinforce these lessons and is similar to online learning applications such as Duolingo for language learning [60]. Our teaching approach is motivated by literature showing that humans learn through examples and employ a nearest neighbor type of mechanism to make decisions on future examples [12, 68, 29]. We follow this literature by showing concrete data examples in the belief that it effectively teaches humans how to interact with AI. We improve on the approach in [57] by incorporating the introductory phase, showing pre-defined region descriptions more clearly established, and iterating on misclassified examples.

**Recommendations in AI Dashboard.** At test time, we can check whether an example $x$ and its corresponding AI output $a$ fall into one of the learned regions $N_k$. If they do, then our dashboard shows the associated recommendation $r_k$ and description $t_k$ alongside the AI output $a$.

## 6 Method Evaluation

**Objective.** In this experimental section[2], we evaluate the ability of our algorithms to achieve three aims: (Aim 1) is learning an integration function that leads to a human-AI team with low error; (Aim 2) is discovering regions of the data space that correspond to the underlying regions where human vs AI performance is different; and, (Aim 3) is evaluating the ability of our region description algorithm to come up with accurate descriptions of the underlying regions. Full experimental details are in Appendix E.

**Datasets and AI Models.** The experiments are performed on two image object detection datasets and two text datasets. The image datasets include Berkeley Deep Drive (BDD) [83] where the task is to detect the presence of traffic lights from blurred images of the validation dataset (10k), and the validation set of MS-COCO (5k) where the task whether a person is present in the image [3] [48]. The text-based validation datasets comprise of Massive Multi-task Language Understanding (MMLU) [33], and Dynamic Sentiment Analysis Dataset (DynaSent) [61]. The pre-trained Faster R-CNN models [65] are considered for the BDD and MS-COCO. For MMLU, a pre-trained flan-t5 model [20] is utilized, whereas a pre-trained sentiment analysis roBERTa-base model is used for DynaSent [8]. Each dataset is split into 70-30 ratio for training and testing five different times so as to obtain error bars of predictions. We obtain embeddings for the text datasets using a sentence transformer [64] and CLIP for the image datasets [62]

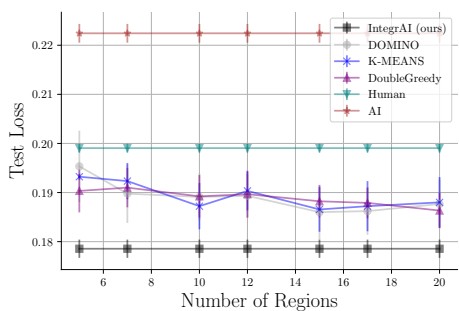

Figure 2: Test Error (↓) of the human-AI system when following the decisions of the different integrators baselines as we vary the number of regions maximally allowed for each integrator on the BDD dataset.

**Baselines.** We benchmark our algorithm with different baseline methods that can find regions of the space. The baselines include: (a) DOMINO [24] which is a slice-discovery method for AI errors, (b) K-Means following the approach of [63], and (c) the double-greedy algorithm from

---

[2]Code is available in https://github.com/clinicalml/onboarding_human_ai.

[3]We extend this to detecting presence of any object.

[57] that finds regions for Human-AI onboarding. For the regions obtained from these baselines, we compute the optimal integration decision that results in minimal loss. For our method, we set $\beta_u = 0.5, \beta_l = 0.01, \alpha = 0.0$ for Aim 1 and $\beta_u = 0.1, \beta_l = 0.01, \alpha = 0.5$ for Aim 2 and random prior decisions (50-50 for 0 and 1). In the context of the region-description algorithms, we compare to the SEAL approach [63], a simple baseline that picks the best representative description from the existing dataset (best-caption) and ablations of our method.

Table 1: Error ($\downarrow$) on the test set (in %) (region discovery) of the human-AI system when following the integrator resulting from the different methods at region size 10 on the different non-synthetic datasets.

|  | BDD | MMLU | DynaSent | MS-COCO |
|---|---|---|---|---|
| IntegrAI (ours) | **17.8 $\pm$ 0.2** | **45.3 $\pm$ 0.3** | 20.2 $\pm$ 0.3 | **22.6 $\pm$ 0.4** |
| DOMINO [24] | 18.9 $\pm$ 0.4 | 48.1 $\pm$ 0.2 | 20.0 $\pm$ 0.2 | 22.7 $\pm$ 0.4 |
| K-MEANS [63] | 19.0 $\pm$ 0.5 | **45.3 $\pm$ 0.3** | 20.0 $\pm$ 0.2 | 23.2 $\pm$ 0.1 |
| DoubleGreedy [57] | 18.9 $\pm$ 0.1 | 46.1 $\pm$ 0.6 | 20.0 $\pm$ 0.2 | 23.8 $\pm$ 0.4 |

Table 2: Clustering metrics (Adjusted Rand index [69] $\uparrow$, Fowlkes–Mallows index [26] $\uparrow$) of the regions (10 regions) found by the different methods on the synthetic dataset setup.

|  | BDD | MMLU | MS-COCO |
|---|---|---|---|
| IntegrAI (ours) | **(0.06,0.40)** | **(0.41, 0.84)** | (0.08,**0.53**) |
| DOMINO [24] | (0.02,0.25) | (0.17,0.65) | (0.04,0.43) |
| K-MEANS [63] | (0.04,0.27) | (0.08,0.49) | **(0.08**,0.49) |
| DoubleGreedy [57] | (0.02,0.26) | (0.10,0.52) | (0.02,0.40) |

**Learning Accurate Integrators ( Aim 1).** The goal is to measure the ability of our method in learning integration functions that lead to low Human-AI team error (the loss $L(H, M)$). This can be well represented by measuring the errors on the training set (discovering regions of error) and the test set (generalization ability). In Table 1, we show the results of our method and the baselines at learning integrators and find that our method can find regions that are more informative with respect to Human vs AI performance on the test data and significantly better on the training data. Figure 2 shows that on BDD our method can find an integrator that leads to lower loss at test times than the baselines with a minimal number of regions.

**Recovering Ground truth Regions (Aim 2).** We just established that the regions discovered by our algorithm result in a Human-AI team with lower error than human or AI alone. However, it still needs to be verified if the regions denote any meaningful and consistent regions of space. We utilize a synthetic setup by simulating the AI model and the human responses such that there exist (randomized) regions in the data space where either the human or the AI are accurate/inaccurate. These regions are defined in terms of metadata. As an example on the BDD dataset, we can define the AI to be good at daytime images and bad at images of highways, whereas the human to be good at nighttime images and bad at images of city streets. We employ our algorithm and the baselines to discover the regions and compare them with the ground truth region corresponding to the partition of the data, which is essentially a clustering task with ground truth clusters. Results are shown in Table 2 indicating that the ground truth regions can be recovered to an extent.

**Describing Regions (Aim 3).** We conduct an ablation study where we evaluate the power of contrasting and self-correcting ability of Algorithm 1 against baselines. On the MS-COCO dataset, we take regions defined in terms of the presence of a singular object (e.g., 'apple') and try to uncover a single-word description of the region from the image captions. We use standard captioning metrics that compare descriptions from the algorithms to the object name, we include a metric called "sent-sim" that simply measures cosine similarity with respect to a sentence transformer [64]. We compare to ablations of Algorithm 1 with $m \in \{0, 5, 10\}$ (rounds of iteration) and without having examples outside the region (IntegrAI, $S^- = \emptyset$). Results are in Table 3 and show that including examples outside the region improves all metrics while increasing iterations ($m$) only slightly improves results.

Table 3: Evaluation of our region description algorithm (Algorithm 1) on a synthetic evaluation of MS-COCO where the different algorithms try to describe a set of points that contain the presence of an object. For example, a region is defined if it contains the object "apple". Then the different algorithms try to describe the region and we compare it to the description "apple".

|  | best-caption | SEAL | IntegrAI (m=5) | IntegrAI (m=10) | IntegrAI ($S^- = \emptyset$) | IntegrAI (m=0) |
|---|---|---|---|---|---|---|
| METEOR | 12.9 $\pm$ 1.9 | 9.16 $\pm$ 1.89 | **26.1 $\pm$ 3.3** | 25.4 $\pm$ 3.3 | 24.3 $\pm$ 3.3 | 25.4 $\pm$ 3.2 |
| sent-sim | 39.8 $\pm$ 1.9 | 44.1 $\pm$ 2.5 | 66.0 $\pm$ 3.2 | **68.0 $\pm$ 3.3** | 65.1 $\pm$ 3.2 | 67.0 $\pm$ 3.1 |
| ROUGE | 5.81 $\pm$ 1.2 | 0.0 $\pm$ 0.0 | 27.9 $\pm$ 5.1 | **35.6 $\pm$ 5.5** | 25.6 $\pm$ 4.9 | 32.6 $\pm$ 5.4 |
| SPICE | 12.7 $\pm$ 1.9 | 7.53 $\pm$ 2.3 | **45.2 $\pm$ 5.8** | **45.2 $\pm$ 5.8** | 41.1 $\pm$ 5.8 | 43.8 $\pm$ 5.8 |

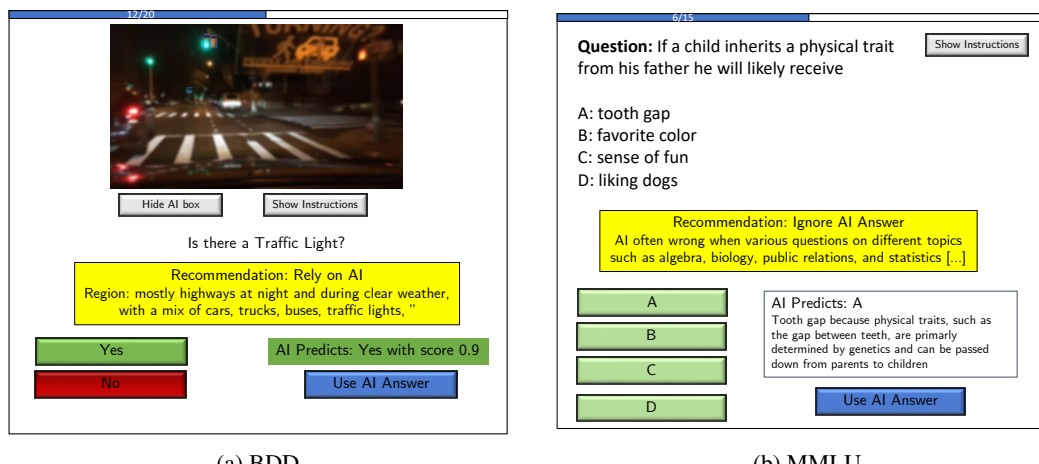

| (a) BDD | (b) MMLU |

Figure 3: (a) Interface for humans to predict a traffic light on images from BDD dataset in the presence of AI's prediction, confidence score, and bounding box and (b) interface for humans to answer multiple choice questions from MMLU dataset with AI's prediction and explanation.

For the apple example, the ablated method finds the description to be "fruit" whereas our method finds it as "apple", thereby eliminating any possibility of confounding effects. More details are provided in the appendix.

## 7 User Studies to Evaluate Onboarding Effect

**Tasks.** We perform user studies on two tasks: 1) predicting the presence of a traffic light in road images from the BDD dataset [83] and 2) answering multiple-choice questions from the MMLU [33] dataset. For BDD, we blur the images with Gaussian blur to make them difficult for humans and use the faster r-cnn model for the AI. Participants can see the AI's prediction, bounding box on the image as an explanation and the model's confidence score. For MMLU, participants are shown a question, four possible answers and have to pick the best answer and the prediction of GPT-3.5-turbo [59]. We also obtain an explanation by using the prompt "Please explain your answer in one sentence", both the AI answer and the explanation are shown. GPT-3.5 obtains an accuracy of 69% during our evaluation and we restrict our attention to specific subjects within the MMLU dataset. Specifically, we sample 5 subjects (out of the 57 in MMLU) where ChatGPT has significantly better performance than average, 5 where it's significantly worse, and 4 subjects where performance is similar to average performance. We sample 150 questions from each subject and additionally sample 150 questions from OpenBookQA dataset [54] to use as attention checks. We show the prediction interfaces in Figure 3. Details are in Appendix F.

**Participants.** We submitted an IRB application and the IRB declared it exempt as is, all participants agreed to a consent form for sharing study data. We recruited participants from the crowdsourcing website Prolific [1] from an international pool filtering for those fluent in English, have above 98% approval rating, have more than 60 previous submissions, and have not completed any of our studies before. For BDD, participants are compensated $3 per 20 examples in the study and then some receive a bonus of $2 for good performance. For MMLU, we pay participants $3 for every 15 questions. We collected information about participants' age, gender (52% identify as Female), knowledge of AI, and other task-specific questions. Participants have to correctly answer three initial images without blur (questions from OpenBookQA for MMLU) and encounter attention checks throughout their studies to further filter them, we exclude participants who fail all attention checks.

**Experimental Conditions.** For BDD, we initially collect responses from 25 participants that predict without the AI and then predict with the help of AI (but no onboarding), we use this data to collect the dataset $D$ of prior human integration decisions and human predictions to find 10 regions using IntegrAI. We then run four different experimental conditions with 50 unique participants in each where they predict on 20 examples: (1) human predicts alone (H) and human predicts with the help of AI (H-AI) but no onboarding and random order between with and without AI, (2) human

| Metric | AI only | Human | Human+AI | Onboard(ours)+Rec | Onboard(ours) | Onboard(baseline) | Rec |
|---|---|---|---|---|---|---|---|
| Accuracy (%) | $79.0 \pm 0.7$ | $78.5 \pm 1.7$ | $77.2 \pm 1.4$ | $79.9 \pm 1.4$ | $82.6 \pm 1.3$ | $80.4 \pm 1.4$ | $81.4 \pm 1.8$ |
| Test vs H-AI | $0.272, 1.211$ | $0.455, 0.752$ | N/A | $0.268, 1.352$ | $0.042, 2.747$ | $0.261, 1.525$ | $0.206, 1.839$ |
| AI reliance (%) | N/A | N/A | $16.5 \pm 3.1$ | $66.5 \pm 2.4$ | $25.5 \pm 3.4$ | $24.4 \pm 4.4$ | $21.4 \pm 3.2$ |
| Time/example (s) | N/A | $5.408 \pm 0.289$ | $7.78 \pm 0.517$ | $7.622 \pm 0.371$ | $5.936 \pm 0.288$ | $6.841 \pm 0.543$ | $8.717 \pm 0.516$ |

Table 4: Results from our user studies. For accuracy, time per example, and AI-reliance we report mean and standard error across participants. The "Test vs H-AI" row reports the adjusted p-value and t-test for a two-sample t-test between the human+AI condition and the other conditions (columns).

| Metric | AI only | Human | Human+AI | Onboard(ours)+Rec | Onboard(ours) | Rec |
|---|---|---|---|---|---|---|
| Accuracy (%) | $72.9 \pm 0.6$ | $52.8 \pm 2.2$ | $75.0 \pm 1.7$ | $73.7 \pm 1.8$ | $74.4 \pm 1.7$ | $69.8 \pm 1.8$ |
| Test vs H-AI | $0.230, -1.488$ | $0.0, -7.899$ | N/A | $0.747, -0.53$ | $0.792, -0.265$ | $0.101, -2.08$ |
| AI reliance (%) | N/A | N/A | $40.0 \pm 3.6$ | $34.5 \pm 3.7$ | $40.6 \pm 3.8$ | $34.2 \pm 2.9$ |
| Time/example (s) | N/A | $30.608 \pm 2.109$ | $23.623 \pm 1.66$ | $22.917 \pm 1.362$ | $20.977 \pm 1.509$ | $29.535 \pm 1.883$ |

Table 5: Results from our user studies for MMLU.

receives onboarding using our method and then in a random order also receives recommendations (Onboard(ours)+Rec) or no recommendations (Onboard(ours)), (3) human goes through only a modified onboarding procedure that only uses step 1 and step 2 from section 5 and uses regions from DOMINO [24] (Onboard(baseline)) and finally (4) human does not receive onboarding but receives the AI-integration recommendations (Rec). For MMLU, participants are tested on 15 examples per condition and onboarding goes through 7 regions found by our algorithm. We run `IntegrAI` on both the dataset embeddings, metadata (subject name), and an embedding of ChatGPT explanations separately. We find 10 regions based on the metadata and 2 regions based on the ChatGPT explanations. Due to budget constraints, we only run conditions 1-2-4. Note that all participants receive the introduction phase of the onboarding (AI model information) regardless if they go through onboarding or not.

**Results.** In Table 4 and Table 5 we display various results from the user studies for BDD and MMLU respectively across all experimental conditions. We note the average accuracy ($\pm$ standard error) across all participants for their final prediction, how often they pressed the "Use AI Answer" button that we call AI-reliance, the time it took for them to make a prediction on average per example (we filter any time period of more than 2 minutes). Finally, we compute using a two-sample independent t-test (* a paired t-test only between Human and Human-AI conditions) the p-value and t-test when comparing each condition (the columns) to the Human-AI condition where the human receives no onboarding. Since we perform multiple tests, we need to correct for multiple hypothesis testing so we rely on the Benjamini/Hochberg method.

**Analysis.** For BDD, we first observe that human and AI performance is very comparable at around 79%, which reduces slightly to 77.2% when the human collaborates with the AI without onboarding. Participants who go through onboarding have a significantly higher task accuracy compared to those who didn't go through onboarding (corrected p-value of 0.042) with a 5.4% increase. The onboarding baseline fails to significantly increase task accuracy, showcasing that the increase is not due to just task familiarity but possibly due to insights gained from regions showcased. Displaying recommendations in addition to onboarding (Onboard(ours)+Rec) does not improve performance but adds time to the decision-making process (7.6s compared to 5.9s without). For MMLU, we note that there is a 20% gap between human and AI performance, but human+AI with and without onboarding can obtain an accuracy of around 75% which is slightly higher than AI alone; onboarding had no additional effect. Interestingly, we find a weakly significant negative effect of only showing AI-integration recommendations which decrease accuracy by 5% and adds 6 seconds of time per example.

**Discussion.** We believe that for MMLU, due to the wide gap between human and AI accuracy and convincing ChatGPT explanations, onboarding did not improve performance. Moreover, it is clear that in their current form, displaying the AI-integration recommendation is not an effective strategy and that onboarding on its own is sufficient. Finally, note that even the Human-AI baseline benefits from the human-AI card which might explain that the team is at least as good as it's components.

**Limitations.** Onboarding and recommendations can significantly affect human decision making. If the recommendations are inaccurate, they could lead to drops in performance and thus require safeguarding. Onboarding and recommendations can be tailored to the specific human by leveraging their characteristics to few-shot learn their prior integrator and prediction abilities.

## Acknowledgments

We thank Hunter Lang and Arvind Satyanarayan for feedback on early stages of this work. HM is thankful for the support of the MIT-IBM Watson AI Lab.

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

# Part I

# Appendix

## Table of Contents

## A   Extended Related Work

Reference [47] proposes a method for human-AI collaboration via conditional delegation rules that the human can write down. Our framework enables the automated learning of such conditional delegation rules for more general forms of data that can also depend on the AI output. [77] proposes to modify the confidence displayed by the AI model to appropriately encourage and discourage reliance on the AI model. However, this technique deliberately misleads the human on the AI model ability, our methodology incorporates similar ideas by learning the human prior function of reliance on the AI and then improving on it with the learned integration recommendations, however, we display these recommendations in a separate dashboard without modifying the AI model output. A related approach to our methodology by [51] is to adaptively display or hide the AI model prediction and display the estimated confidence level of the human and the AI on a task of predicting whether a person's income exceeds a certain level. They show that displaying the confidence of the human and the AI to the human improves performance. Our method is able to learn the confidence level of the human and the AI, but also incorporates how the human utilizes the AI and describes the regions where AI vs human performance is different. [16] presents a similar approach to our AI recommendations, however, they use simulated and faked AI models and descriptions of behavior while we are able to obtain automated generation of these descriptions of AI behavior.

Existing research has examined various methods to establish human trust in and replicate the predictions of machine learning models. One such method is LIME, a black-box feature importance technique, which was employed to select examples for evaluation by crowdworkers to determine the superior model among two options [67, 45]. However, their selection strategy disregards the human predictor, and their approach merely presents the examples without further action. In the context of visual question answering, Chandrasekaran et al. [18] manually selected seven examples to educate crowdworkers about the AI's capabilities, leading to an enhanced ability to identify instances where the AI failed. Feng et al. [25], in the domain of Quizbowl question answering, emphasize the significance of incorporating the human expert's skill level when designing explanations. This further justifies our decision to involve the human predictor in the selection of teaching examples. Cai et al. [17] conducted a study involving 21 pathologists to gather guidelines on what clinicians desired to know about an AI system prior to interacting with it. Yin et al. [84] investigated the impact of initial debriefing on stated AI accuracy versus observed AI accuracy during deployment, finding a substantial influence of stated accuracy on trust that diminishes quickly once the model is observed in practical use. This reinforces our approach of building trust through examples that

simulate real-world deployment. Bansal et al. [5] examined the role of the human's mental model of the AI in task accuracy; however, the mental model was developed through interaction during testing rather than during an initial onboarding stage. The most similar work to ours is that of [57] which presents an onboarding scheme based on selecting a set of examples and allows the human to describe the regions where the AI performance is good or bad. Through a user study on passage-based question answering, they show that their onboarding scheme improves performance by 5%, however, they evaluate without the presence of AI evaluations, with a synthetic AI model and their scheme involves more involvement from the human as they have to describe the regions themselves. Another approach to teaching involves providing humans with guidelines on when to rely on AI systems [3]. Model cards [55] and industry practices such as the IBM AI fact sheet [4] demonstrate direct methods of presenting these guidelines to users, we present humans with a similar form of card but that includes aspects of human performance "human-AI card".

There is a growing and large area of literature on discovering (and auditing) regions of AI error, the following is not meant as an extensive list of related work but captures some of the essence of the literature:

- Adatest allows a user to iteratively discover regions of AI error using LLMs for NLP tasks and then re-train the model on the regions of error [66], it was then extended to vision tasks [27] with a similar procedure in [80].

- Erudite allows users to discover regions of error of NLP models through user interfaces [82], there is a wider literature on dashboards for discovering regions of error [2]

- [37] learns an SVM model from image embeddings to predict model error and then uncover regions of error based on the directions of the SVM model.

- Works have done extensive manual annotation of ImageNet model mistakes[35, 76].

- DOMINO discovers regions of model error using a slice discovery model based on a specialized gaussian mixture model [24], extensions include DRML [87].

- The Spotlight method learns individual regions based on a neighborhood of a learned point [23], our region finding algorithms generalize their procedure by learning weighted distance distance measures and with a different aim of improving gain of the prior.

- SEAL: Interactive Tool for Systematic Error Analysis and Labeling, uses k-means to uncover regions of error and then uses an LLM to describe each region [63].

Recent work has emerged on describing sets of images [34, 53] but they don't incorporate a contrastive method as we propose. Helpful tools for describing differences between text and images can be useful for describing regions which future work can incorporate [89, 79, 78].

One of the objectives of explainable machine learning is to enhance humans' ability to assess the accuracy of AI predictions by offering supporting evidence [44, 43, 70, 32, 88, 42, 72, 73, 81, 31]. Nevertheless, these explanations fail to provide guidance to decision makers on how to balance their own predictions against those of the AI or how to integrate the AI's evidence into their final decision [39]. [75] shows that AI explanations can reduce overreliance and improve human-AI team performance, however, their experiments are with simulated AI models and explanations. The central question of our work is not to study the utility of AI explanations, in fact, all our user studies incorporate AI explanations and we aim to improve human-AI performance in their presence.

## B  Region Finding Algorithm - Details

**Regions Requirements.**  Each region in our algorithm should aim to satisfy the following constraints:

1. **Region Size:** We want the size of the region to be at least of size $\beta_l$ and at most of size $\beta_u$.

2. **Consistency of takeaway:** The examples in each region must agree on what the takeaway is in terms of the integration decision. Specifically, at least $\alpha\%$ of all points in the region must either be: ignore AI, use AI as is or integrate AI advice.

3. **Concise and Distinguishable Theme:** Each region must be concisely described in natural language in such a way to differentiate from the overall domain. If a region cannot

be described in natural language, the human may not be able to derive a generalizable recommendation from it. This is a constraint that we implicitly try to satisfy by learning neighborhoods in a natural language embedding space.

4. **Minimum Gain:** Each region must have a minimum information gain (defined below) of $\delta$. This is to ensure that all regions contain sufficient novel information to the human.

The optimization to find each region can be formulated in its non-relaxed form as:

$$\max_{c,\gamma,w,r} \sum_{i=1}^{n} \mathbb{I}_{||w((e_i,a_i)-c)||<\gamma} \cdot \boldsymbol{g}_{i,r}$$

$$\text{s.t.} \quad \sum_{i=1}^{n} \mathbb{I}_{||w((e_i,a_i)-c)||<\gamma} \cdot \mathbb{I}_{r_i=r} \geq \alpha n$$

$$\text{s.t.} \quad n\beta_l \leq \sum_{i=1}^{n} \mathbb{I}_{||w((e_i,a_i)-c)||<\gamma} \leq n\beta_u$$

And the relaxation we propose is (refer back to Section 4.1):

$$\max_{c,\gamma,w,r} \sum_{i=1}^{n} \sigma(C_1(-||w \circ ((e_i,a_i)-c)|| + \gamma)) \cdot g_{i,r} - \lambda \max(\sum_{i=1}^{n} \sigma(C_1(-||w \circ ((e_i,a_i)-c)|| + \gamma))$$

$$\cdot (\mathbb{I}_{r_i^*=r} - \alpha n), 0) - \lambda \max \left( \sum_{i=1}^{n} \sigma(C_1(-||w \circ ((e_i,a_i)-c)|| + \gamma)) - \beta_u n, 0 \right)$$

$$\lambda \max \left( -\sum_{i=1}^{n} \sigma(C_1(-||w \circ ((e_i,a_i)-c)|| + \gamma)) + \beta_l n, 0 \right)$$

We run the optimization for $r = 0$ and $r = 1$ and choose the $r$ with the better objective value. With $r$ fixed, we optimize with respect to the remaining continuous parameters using AdamW and reduce the learning rate when loss has stopped improving. We initialize with $\gamma = 0$ and $w = \mathbf{1}$. For the centroid $c$, we first run $k$-medoids clustering on the input data with $k = \min(\max(100, T), n)$ and randomly select 20 of the resulting centroids. Then with $c$ initialized as each one of the 20 centroids in turn, we run the optimization for 200 epochs and record the loss, and finally optimize for 2000 epochs with the best initialization for $c$. We do these repeated initializations to avoid local minima which are a common failure mode of this type of optimization problem. Note this process is to find one region, to find all regions we repeat this process identically to find regions one by one.

**Selection Based Approach.** We described in the body of the paper a generative algorithm to find the regions. We now describe a selection based algorithm that finds centroids from points in the dataset $\widetilde{D} = \{e_i, r_i\}$. We will also restrict the radius to be the distance between the centroid to another data point in $\widetilde{D}$. We proceed with a sequential search, at round $i$ we perform the following search:

$$c_i, \gamma_i = \arg \max_{i \in \widetilde{D}, \gamma} G(N_{i,\gamma}, \hat{R}^*, R_{N_{1:i-1}}), \tag{5}$$

$$\text{s.t.} \ \exists k \in [n] \ s.t. \ \gamma = d(e_i, e_k), \tag{6}$$

$$\text{and} \ \frac{\sum_{j \in [n], d(e_i, e_j) < \gamma} \mathbb{I}_{r_j=r_i}}{|\{j \in [n], d(e_i, e_j) < \gamma\}|} \geq \alpha \tag{7}$$

$$\text{and} \ \beta_l \leq |\{j \in [n], d(e_i, e_j) < \gamma\}| \leq \beta_u \tag{8}$$

$$\text{and} \ G(N_{i,\gamma}, \hat{R}^*, R_{N_{1:i-1}}) > \delta \tag{9}$$

Note that with the selection-based procedure, we have to define a fixed distance measure $d$, and we cannot optimize over the deferral decision $r$ of the region as we inherit from the point $i$ found. The naive algorithm to solve the above search is as follows. We first compute the distance matrix between all data points in $\widetilde{D}$, call this matrix $K$. We then at each round do the following for each point $i$ in $D\widetilde{D}$: sort the points by their distance to $i$, iteratively grow the region around the point $i$ to satisfy the

constraints, and then keep track of the maximum gain radius. Each time we grow the region by one point, we check if the constraints are satisfied. The search is parallelized across multiple instances to make it faster. Finally, we compare the gain of all feasible points and pick the highest one. This algorithm extends the approach of [57] to incorporate additional constraints and has significant speed ups over their approach.

**Aggregating Regions Across Different Embedding Spaces**  . We can run our region finding algorithm above and find different regions across multiple embedding spaces. The question is how do we aggregate these regions. Say we found a set of $T$ regions $N_1, \cdots, N_T$. We then run the following meta-algorithm: do a greedy sequential search to add regions one at a time while making sure the minimum gain requirement is satisfied, stop when it is not.

## C   Region Description Algorithm - Details

The LLM call inside Algorithm 1 is accomplished by building the prompt with $S^+$ being the inside set (positives) and $S^-$ being the outside set (negatives) as follows:

```
def get_prompt(positives, negatives):
    prompt = pre_instruction + "\n"
    prompt += "inside the region: \n "
    counter = 1
    for p in positives:
        prompt += p[0] + ", \n "
        counter += 1
    if len(negatives) > 0 :
        prompt += ". \n not in the region: \n"
        counter = 1
        for p in negatives:
            prompt += p[0] + ",\n"
            counter += 1
    prompt += post_instruction
    return prompt
```

For the experiments in Section 6 we use the following pre instruction:

> I will provide you with a set of descriptions of points that belong to a region and a set of descriptions of point that do not belong to the region. Your task is to summarize the points inside the region in a concise and precise short sentence while making sure the summary contrasts to points outside the region. Your one sentence summary should be able to allow a person to distinguish between points inside and outside the region while describing the region well. The summary should not be a single word, it should be accurate, concise, distinguishing, and precise.
> Example:
> Inside the region:
> - two cows and two sheep grazing in a pasture.
> - the sheep is standing near a tree.
> Not in the region:
> - the cows are lying on the grass beside the water.
> summary: sheep.
> End of Example

While the post-instruction is simply "summary:".

For the ablation without contrasting, the pre instruction we use is:

> I will provide you with a set of descriptions of points that belong to a region.Your task is to summarize the points inside the region in a concise and precise short sentence .Your one sentence summary should be able to allow a person to distinguish points inside the region while describing the region well.The summary should be a single word, it should be accurate, concise, distinguishing and precise.

Example:

inside the region: - two cows and two sheep grazing in a pasture.

-the sheep is standing near a tree.

summary: sheep.

End of Example

For the user studies, we had used an earlier instruction that works slightly worse:

"summarize the points inside the region in a concise and precise short sentence while making sure the summary contrasts to points outside the region"

We recommend the use of the following pre instruction:

I will provide you with a set of descriptions of points that belong to a region and a set of descriptions of points that do not belong to the region. Your task is to summarize the points inside the region in a concise and precise short sentence while making sure the summary contrasts to points outside the region. Your one sentence summary should be able to allow a person to distinguish between points inside and outside the region while describing the region well. The summary should be no more than 20 words, it should be accurate, concise, distinguishing and precise.

Example:

inside the region:

- two cows and two sheep grazing in a pasture.

- the sheep is standing near a tree.

outside the region:

- the cows are lying on the grass beside the water.

summary: The region consists of descriptions that have sheep in them outside in nature, it could have cows but must have sheep.

End of Example

## D   Onboarding and Recommendations to Promote Rules - Details

**Introductory Phase.**   To facilitate a smooth onboarding process for individuals working with an AI assistant, we introduce the Human-AI Card. This card provides detailed insights into the AI's capabilities, training, and performance. Below is a presentation of the core details of the card inspired by the work in [17]:

Table 6: Human-AI-Card presented to the human as part of onboarding

| Information | Description |
| --- | --- |
| AI Input | What the AI uses to make its prediction |
| AI Output | What the AI provides as output (predictions, explanations, ...) |
| Source of Training Data for AI | Description of data used to train the AI |
| Source of Pre-Training Data of AI | Description of pre-training data of model AI is based on |
| Training Objective of AI | what the AI is trying to achieve (minimize classification error, detect objects, next word prediction) |
| Average AI Performance | |
| Average Human Performance | |

Additionally, we provide a breakdown of AI and Human performance on different subgroups of data.

We take an example of the Berkeley Deep Driving dataset, where a subgroup might comprise images taken during the night, in rainy weather, or on a highway. We compute the model's error for each possible subgroup and then perform a paired t-test comparing the subgroup model error to the average model error over the entire data. For the purpose of our user studies, we highlight subgroups

defined by a single metadata category that show statistically significant differences ($p \leq 0.05$). It's important to note that, for rigorous analysis, one should apply corrections for multiple hypothesis testing. However, considering the vast number of metadata categories, many results might become insignificant. Therefore, for simplicity, we adopt this heuristic approach.

# E  Method Evaluation - Details

In Table 7 we report the details on the datasets we use in our method evaluation. We normalize all datasets using $l_\infty$ normalization and run our algorithms for 2000 epochs with a learning rate of 0.001 using AdamW. We use a constant $C_1 = 20$, $\delta = 1$, $\beta_u = 0.1$ ($\beta_u = 0.5$ for Aim 1), $\alpha = 0.5$ and $\beta_l = 0.01$ for all experiments. We use a random prior $R_0$ that has equal probability for 1 and 0 uniformly at random. All experiments are run on a GeForce GTX 1080 Ti.

Table 7: Datasets for "Learning Accurate Integrators ( Aim 1)". We note the total number of samples $n$, the target set size $|\mathcal{Y}|$, the human expert finally the model of the AI. When we note human "XX% accurate", this indicates a synthetic human model that is accurate uniformly at random with probability XX%. For DynaSent, the AI model is a a pre-trained sentiment analysis roBERTa-base model [8] on Twitter data and achieves 75% accuracy. For both BDD and MS-COCO we blur the images using a Guassian Blur with scale 21 and variance 5.

| **Dataset** | $n$ | $|\mathcal{Y}|$ | **Human** | **AI** |
|---|---|---|---|---|
| Berkeley Deep Drive (BDD) [83, 85] | 10k | 2 | 80% accurate | faster rcnn r50 fpn 1x [4] - Gaussian blur with scale 21 and variance 5 |
| MS-COCO [48] | 5k | 2 (presence of person in image) | 70% accurate | faster rcnn R 50 FPN |
| Massive Multi-task Language Understanding (MMLU) [33] | 14k | 4 (MCQ) | 50% accurate | flan-t5xl [20] |
| Dynamic Sentiment Analysis Dataset (DynaSent) [61] | 6.5k | 3 | leave-one-out annotator | a pre-trained sentiment analysis roBERTa-base model [8] |

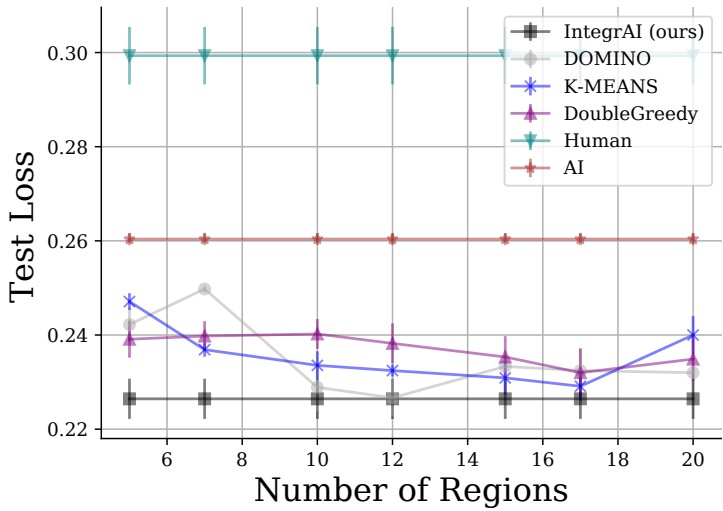

Figure 4: Test Error (↓) of the human-AI system when following the decisions of the different integrators baselines as we vary the number of regions maximally allowed for each integrator on the MS-COCO dataset.

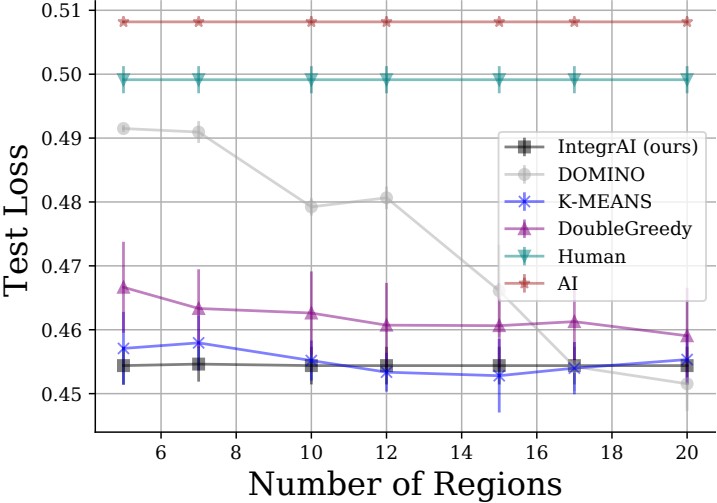

Figure 5: Test Error (↓) of the human-AI system when following the decisions of the different integrators baselines as we vary the number of regions maximally allowed for each integrator on the MMLU dataset.

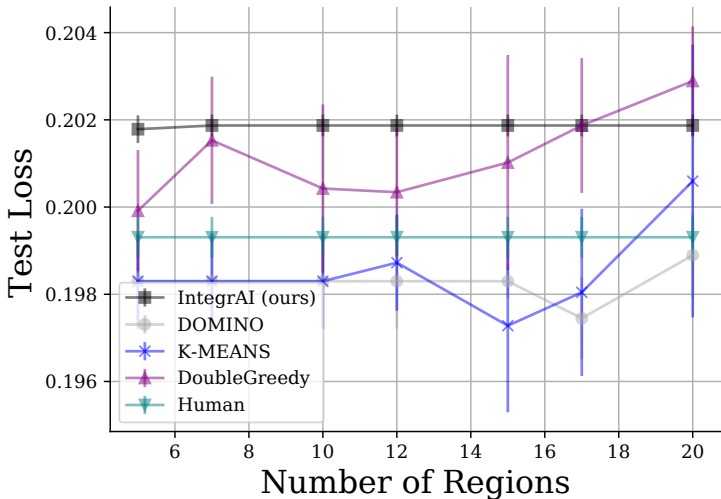

Figure 6: Test Error (↓) of the human-AI system when following the decisions of the different integrators baselines as we vary the number of regions maximally allowed for each integrator on the Dyanasent dataset.

For Aim 2, we create synthetic AI and human models as follows:

- BDD: AI and Human model each have non-overlapping two to four regions defined as the intersection of two metadata categories, each region is either a good region where the AI/Human have 95% accuracy or bad region with 60% accuracy (equal number of good and bad regions). If an example does not belong to any region the AI/Human have 75% accuracy. Each region is of size at least $0.01$ and at most $0.2$ in terms of fraction of data points in the dataset. An example region is "AI is good at: weather: clear and scene: highway" or "Human is bad at: weather: snowy and timeofday: night ".

- MMLU: same setup as BDD, but there is only one metadata category in the dataset, so each region belongs to a certain metadata category. An example region is "AI is good at: subject: professional psychology".

- MS-COCO: same setup as BDD. An example region is "'Human is bad at: cow: present".

For Aim 3, an obstacle to quantitative results for region descriptions is that they are often complex even when regions are synthetically defined in terms of metadata and captioning metrics are not informative. For a region defined on BDD as images of "scenes of highways during the night with no cars", our algorithm finds the description "highway during the night with various weather conditions and not congested with many cars" while the SEAL [63] describes it as "Highway Nighttime Weather Conditions". The SEAL description surprisingly has higher captioning metric scores (BLEU, METEOR) [50] while being less informative.

For the MS-COCO evaluation, we only select objects that have at least 50 examples present in the evaluation set which leads to only 73 objects over which we evaluate the different region description algorithms.

# F   User Studies - Details

## F.1   BDD Study

**Task.**   The images from BDD are blurred using a Gaussian Blur with a scale of 21 and a variance of 5. The AI model is a trained faster rcnn model that achieves 84% accuracy without blurring which decreases to 78% accuracy on the test set. We use the bounding boxes from the model and output them on the image (allowing the user to either hide or show them). To get a confidence score, we take the maximum score for the prediction of a traffic light in the image.

**Initial Data Collection.** The BDD dataset was split 70-30 where the 70% split was used to get the initial human predictions and find the regions and the 30% split was used only to get testing examples for the final user study. As mentioned, we obtained data on 400 examples with both human predictions and prior AI-integration decisions. We use these examples to build Random Forrest models that predict both the human predictions and AI-integration decisions from the embeddings, labels, and AI predictions (AI predictions only for predicting AI integration decisions). We use these predictions from the RF models to label the entire dataset. We ensure that the predictions of the RF models are calibrated (if the human is 80% accurate, the model is also 80% accurate) with the human predictions and integration decisions by modifying the threshold on model probability used to make predictions (e.g. from the usual 0.5 threshold to the value that makes the models calibrated). Each participant is evaluated on a randomized set of examples, we create 40 different sets of 20 images that get assigned randomly to each participant.

**Attention Checks.** In each condition, we insert 3 attention checks for every 20 examples where the images are unblurred and we keep the AI prediction. We only retain responses where participants don't get all attention checks wrong. The class balance of the dataset is close to 51%-49%. The attention checks are used as part of the study results as they only modify the blur of the images.

**Regions Found.** The regions found by our procedure are shown in Table 8.

Table 8: Region descriptions found by `IntegrAI` for the BDD user study.

| Region ID | Description |
|---|---|
| 1 | depict various city streets and highways during the daytime with different weather conditions, containing pedestrians, cars, trucks, traffic lights, and signs., |
| 2 | depict various types of roads and streets during the daytime with different weather conditions, containing cars, pedestrians, traffic lights and signs, while the outside examples include scenes with fewer objects or in less common locations such as a parking lot., |
| 3 | depict various scenarios of streets and highways with moderate to heavy traffic flow during the day or night, with different weather conditions, along with traffic signs and lights, cars, trucks, buses, and pedestrians. |
| 4 | depict various outdoor scenes during the daytime or nighttime, containing multiple cars, traffic lights, and traffic signs |
| 5 | depict various city and residential scenes during the daytime with different weather conditions and contain a variety of vehicles, pedestrians, traffic lights, and signs, while the outside examples depict specific limited scenarios with fewer elements., |
| 6 | depict various traffic scenes, mostly highways at night and during clear weather, with a mix of cars, trucks, buses, traffic lights, and traffic signs |
| 7 | depict various city streets and highways with clear weather and a moderate amount of traffic, including cars, signs, and occasionally pedestrians, bicycles, and trucks |
| 8 | depict various urban and residential scenes during different times of day and weather conditions, with a diverse range of vehicles, pedestrians, traffic signs, and traffic lights present |
| 9 | depict various scenes of city streets and highways with typical traffic conditions and without severe weather conditions |
| 10 | depict various scenes of city streets and highways with varying weather conditions, traffic, and signage. |

## F.2 MMLU Study

**Task.** We rely on the MMLU dataset [33] where participants are shown a question, four possible answers and have to pick the best answer (see screenshots in Figure below for examples). We use ChatGPT, also known as GPT 3.5 turbo as our AI model [59]. We obtain the predictions of ChatGPT

on the MMLU dataset following the approach in the official repo of MMLU [5]. We also ask ChatGPT to explain it's answer by using the prompt "Please explain your answer in one sentence". Both the AI answer and the explanation are shown.

ChatGPT obtains an accuracy of 69% during our evaluation and we restrict our attention to specific subjects within the MMLU dataset. Specifically, we sample 5 subjects (out of the 57 in MMLU) where ChatGPT has significantly better perform performance than average, 5 where it's significantly worse, and 4 subjects where performance is similar to average performance. These subjects are listed here:

> high school government and politics, marketing, high school psychology, logical fallacies, sociology, public relations, high school computer science, anatomy, business ethics, elementary mathematics, high school statistics, machine learning, moral scenarios, global facts

We sample 150 questions from each subject and additionally sample 150 questions from OpenBookQA dataset [54] to use as attention checks as human performance on OpenBookQA is 91%.

We run `IntegrAI` on both the dataset embeddings, metadata (subject name), and an embedding of ChatGPT explanations separately. We find 10 regions based on the metadata and 2 regions based on the ChatGPT explanations. The regions and their descriptions found by our algorithm are reported in Table 9.

Table 9: Region descriptions found by `IntegrAI` for the MMLU user study.

| Region ID | Description |
|---|---|
| 1 | Related to marketing, including pricing strategies, branding, communication, product classification, market segmentation, advertising, and supply chain management. |
| 2 | Questions related to psychology and neuroscience. |
| 3 | Questions related to global statistics and trends, ranging from military spending to mental health disorders. |
| 4 | Questions inside the region cover a variety of topics in the field of public relations, including ethical frameworks, evaluation models, common tactics, and regulations. |
| 5 | Mathematical and quantitative questions, involving calculations and problem-solving. |
| 6 | Questions related to sociology, including topics such as social class, symbolic interactionism, bureaucracy, and globalization. |
| 7 | Questions and descriptions of logical fallacies and syllogisms. |
| 8 | Questions focus on US politics, government, and history. |
| 9 | Contains questions related to anatomy and physiology. |
| 10 | Various topics including ethics, regulation, consumer rights, and corporate transparency. |
| 11 | Various questions on different topics such as algebra, biology, public relations, and statistics with multiple options to choose from. |

## F.3 Screenshots of User Study Interface for BDD

---

[5] https://github.com/hendrycks/test

## Classify Images of Road Scenarios

You are invited to participate in a research study how people classify images. You will be presented with a series of images and have to make judgements. You may additionally be asked to interact with an AI model and write a few sentences of text.

This study will include participants over 18 years of age, who feel comfortable using an online interface to read and answer questions in English and investigate images.

To reiterate, you are eligible for this study if and only if:

- Are able to distinguish images of different types.
- You are comfortable reading/writing in English.
- You have not already completed this survey.
- You are over 18 years of age.
- You have JavaScript enabled in your browser.

### What Will My Participation Involve?

If you decide to participate in this research you will be asked to classify different images, write answers to questions in English and interact with AI models.

### Are There Any Risks To Me?

We don't anticipate any risks from participation in this study greater than normal activity.

### Are There Any Benefits To Me?

There are no direct benefits to you other than monetary compensation.

### How Will My Confidentiality Be Protected?

While there will probably be publications as a result of this study, your name will not be used. Only group characteristics will be published.

Figure 7: Consent Form

## Welcome to our experiment! (please read carefully)

After you fill out the information on this page please click the Start Experiment button to proceed.
The experiment may contain multiple stages that you will be guided through.
Please do not refresh or press back the webpage as you may lose a fraction of your progress.
The experiment will contain attention checks that will not modify the question you are asked, but they will be easier questions of the same format.
**There is a potential bonus of up to 2$ for good performance on the tasks.**

### If you have previously completed this study, you are auto-ineligible to complete it again!

Your answers to the below questions have no impact on your eligibility to perform the task.
Your Prolific IDs will NOT be shared with anyone or used in the study or tied to the information below, they are only collected for matching with the recorded study.
By completing this task, you consent to having your responses (excluding age, gender, eyesight and educational ability) be made publicly available for research and educational purposes.

Prolific ID*:

test111

What is your age? Select most appropriate age range.

35-50

What is your gender?

Prefer not to answer

What is the highest degree or level of school you have completed?

Some college credit, no degree

How would you rate your knowledge about Artificial Intelligence (AI) or Machine Learning?

I know how AI works at a basic level

How would you rate your eyesight? Poor indicates any major visual impairments, Excellent denotes 20/20 vision.

Fair

How would you rate your driving ability? Answer this question in terms of safety of driving.

I very rarely drive or I'm not confident in my ability.

Start Experiment

Figure 8: User Information Collection

## Instructions

- This experiment will be in 3 stages.
- Task: You will be shown images of road scenarios from the point of view of inside a car, you will be asked to respond to check whether in the image there is any traffic light (red or green or yellow)
Your goal in is to be as accurate as you can be.
- Interface: you will have to first select an answer by choosing the corresponding button "Yes" or "No" and then pressing the "Submit" button.
You can track how many examples remain by looking at the progress bar on the top of the webpage after you press next. You will keep trying each example until you get the answer correct

### Example (no need to click, only for viewing):

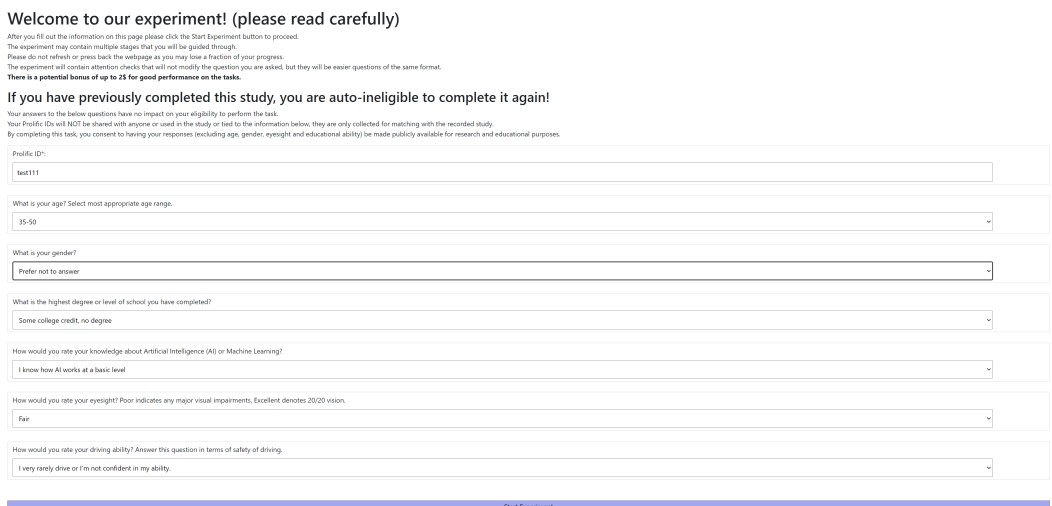

### Is there a traffic light in the picture?

Figure 9: Practice Task instructions

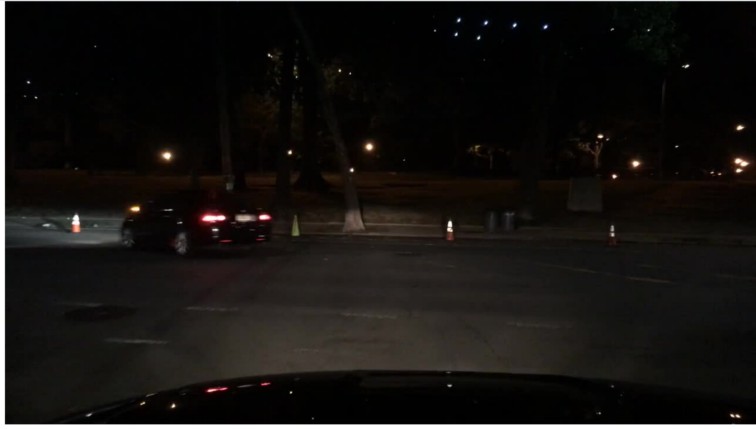

Is there a traffic light in the picture?

Yes

No

Submit

Figure 10: Prediction without AI interface

# Instructions (scroll down for task)

- **AI MODEL:** Images are now blurred and you have access to an Artificial Intelligence (AI) model that provides their prediction
- **AI Button:** A new button now appears called "Use AI Answer" in blue and allows you to select the AI's answer which you can then submit.
    - Please press the button if the AI helped with your answer and you want to use it's answer, if you agree with the answer of the AI but got the answer without it's help then don't press the button.
    - If the AI predicts there to be a traffic light, it will put a box around it in the image and a score (you can hide or show it with a button below the image)
- **Teaching Phase:** The next of examples will help teach you more about the AI through examples where you will receive feedback, after that, there will be a testing phase where the 2$ bonus applies.

Figure 11: Instructions for the onboarding phase

- We give you the following information about the AI model to help you understand when you should use it's answer and when not to:

| | |
|---|---|
| **Average AI Accuracy:** | 78% |
| **Average Human Accuracy:** | 72% |
| **AI Model Input:** | Blurry Image |
| **AI Model Output:** | Prediction of traffic light, bounding box on image showing it's location and a score indicating it's confidence |
| **Source of Training Data:** | Dataset of road images from New York and San Francisco and Bay Area. |
| **AI Training Objective:** | Detect Traffic Lights and other objects in image |

- Now a deeper dive into the AI error/accuracy:

| Category: | Accuracy |
|---|---|
| more than 5 traffic lights in image | 90% |
| If there is 1 to 4 traffic lights | 62% |
| If there is no traffic lights | 86% |
| No Cars | 83% |
| Daytime or overcast weather | 75% |
| few pedestrians | 76% |

Figure 12: Model card information shown during onboarding.

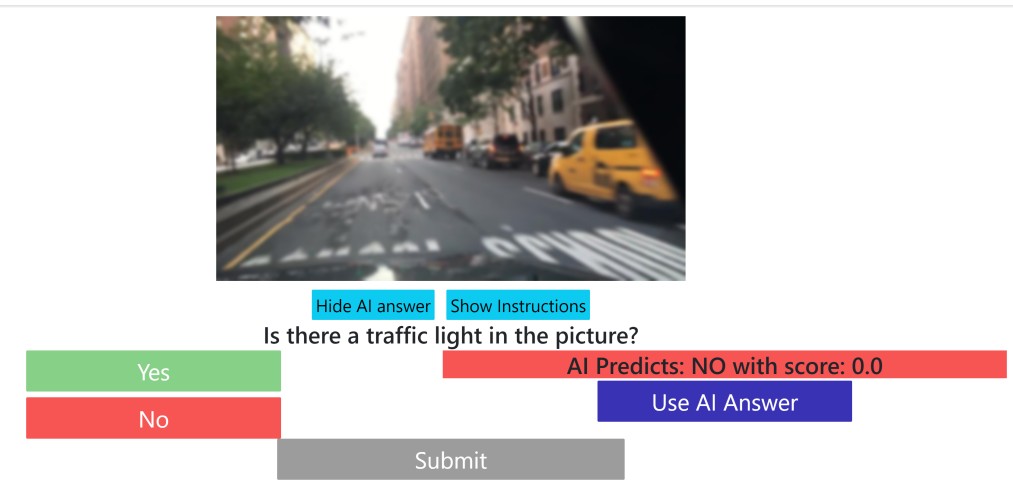

Figure 13: Prediction with AI interface (AI predicts no traffic light)

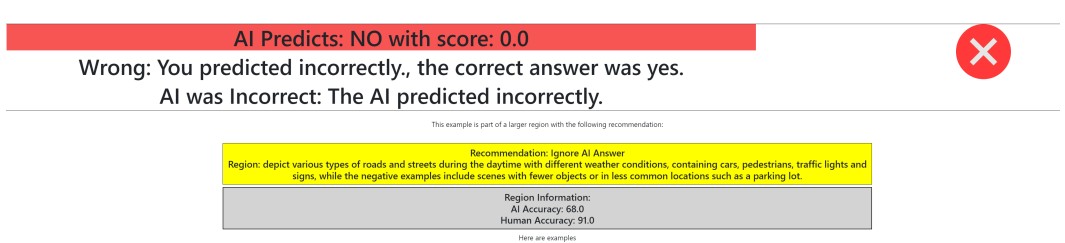

Figure 14: Feedback shown during onboarding phase after human predicts.

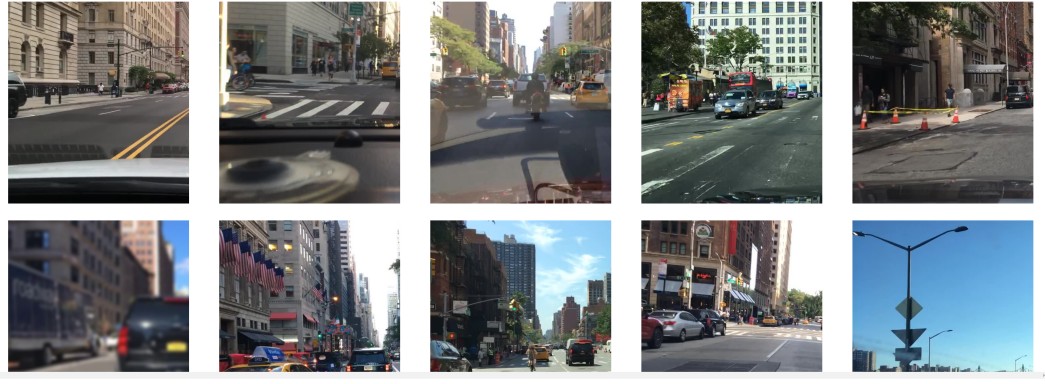

Figure 15: Feedback shown during onboarding phase after human predicts (sets of examples from region)

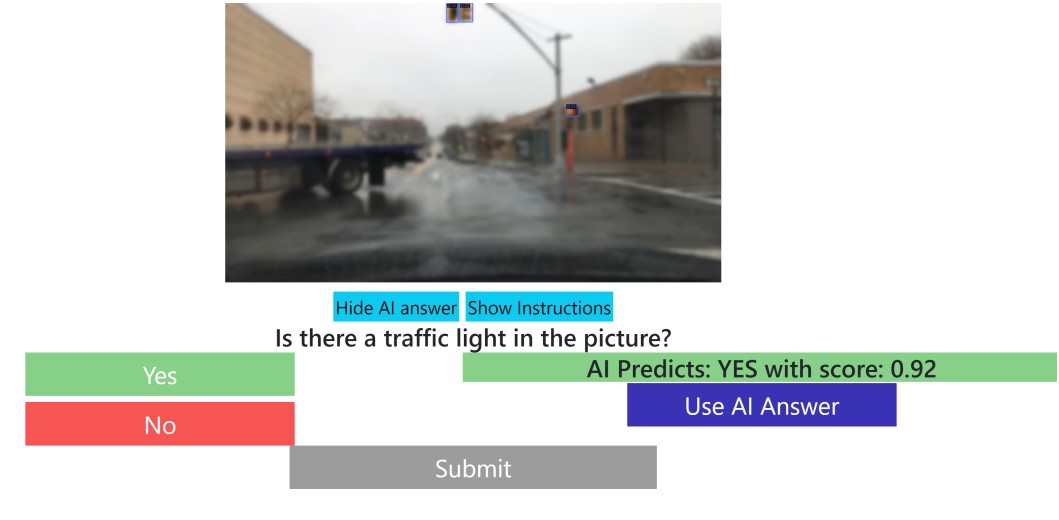

Is there a traffic light in the picture?

Figure 16: Prediction with AI interface (AI predicts there is a traffic light)

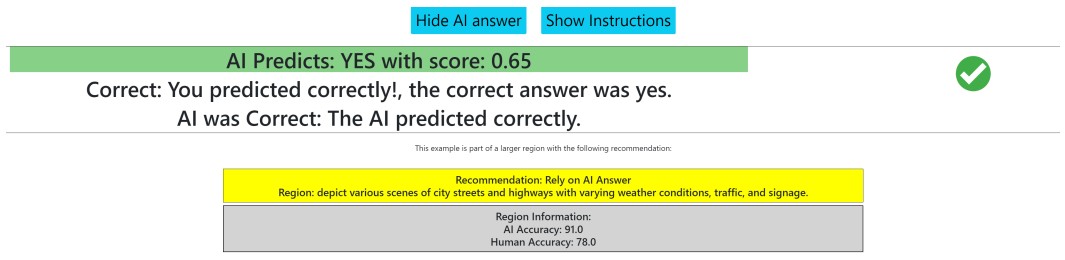

Figure 17: Feedback shown during onboarding phase after human predicts (correct feedback)

Instructions (scroll down for task)

- **Testing Phase:** You will be now tested on a series of examples with the help of the AI model.
- **AI MODEL:** Images are now blurred and you have access to an Artificial Intelligence (AI) model that provides their prediction
- **AI Button:** A new button now appears called "Use AI Answer" in blue and allows you to select the AI's answer which you can then submit.
Please press the button if the AI helped with your answer and you want to use it's answer, if you agree with the answer of the AI but got the answer without it's help then don't press the button.
- If the AI predicts there to be a traffic light, it will put a box around it in the image and a score (you can hide or show it with a button below the image)
Try to do your best to get things correctly as there is a potential bonus of up to 2$.

Please read instructions (25 secs

Figure 18: Testing phase instructions

Instructions (scroll down for task)

- **AI MODEL:** Images are now blurred and you have access to an Artificial Intelligence (AI) model that provides their prediction
- **AI Button:** A new button now appears called "Use AI Answer" in blue and allows you to select the AI's answer which you can then submit.
Please press the button if the AI helped with your answer and you want to use it's answer, if you agree with the answer of the AI but got the answer without it's help then don't press the button.
- If the AI predicts there to be a traffic light, it will put a box around it in the image and a score (you can hide or show it with a button below the image)
- **Our Recommendation:** Additionally, above the button and the AI's answer, we give you a recommendation whether you should use the AI answer or on your own. This recommendation is based on comparing your estimated personal ability compared to the AI's ability, this recommendation is not always correct.
Try to do your best to get things correctly as there is a potential bonus of up to 2$.

Please read instructions (14 secs

Figure 19: Testing phase instructions that include AI-integration recommendation

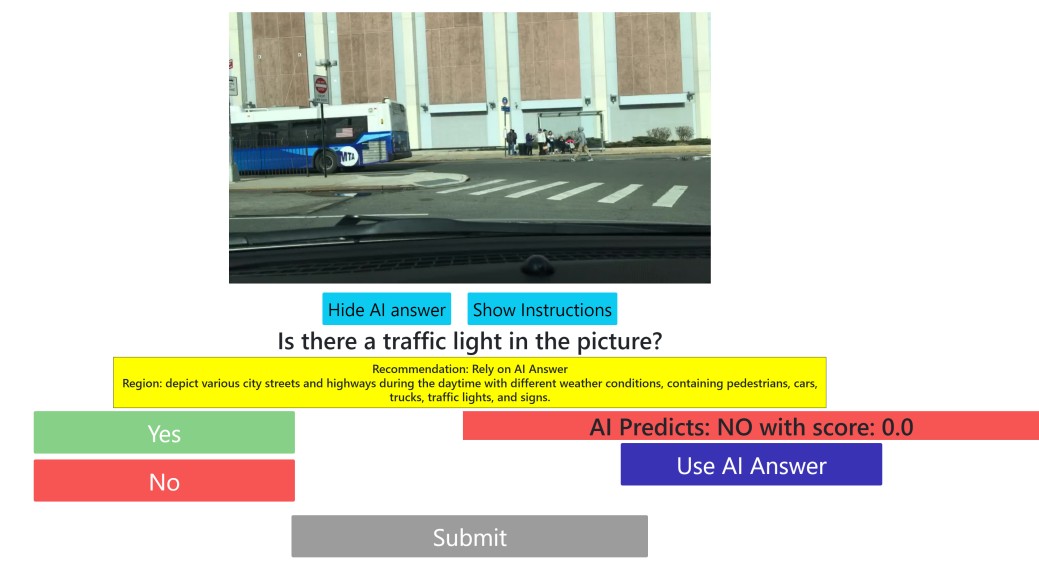

Figure 20: Prediction interface with AI and with AI-integration recommendations.

## F.4   Screenshots of User Study Interface for MMLU

- We give you the following information about the AI model to help you understand when you should use it's answer and when not to:

| | |
|---|---|
| **Average AI Accuracy:** | 69% |
| **Average Human Accuracy:** | 50% (varies between people and subjects) |
| **AI Model Input:** | Question and the answers |
| **AI Model Output:** | An answer and a text explanation |
| **Source of Training Data:** | Trained on all the internet |
| **AI Training Objective:** | Trained to mimic language and respond to instructions |

- Now a deeper dive into the AI error/accuracy:

| **Category:** | Accuracy |
|---|---|
| **Machine Learning** | 43% |
| **Management/Marketing** | 85% |
| **Elementary Math** | 60% |
| **Psychology/Sociology** | 86% |
| **Global Facts** | 42% |
| **Moral Scenarios** | 44% |
| **Logical Fallacies** | 78% |
| **High School Statistics** | 59% |

Figure 21: Model Card for MMLU study

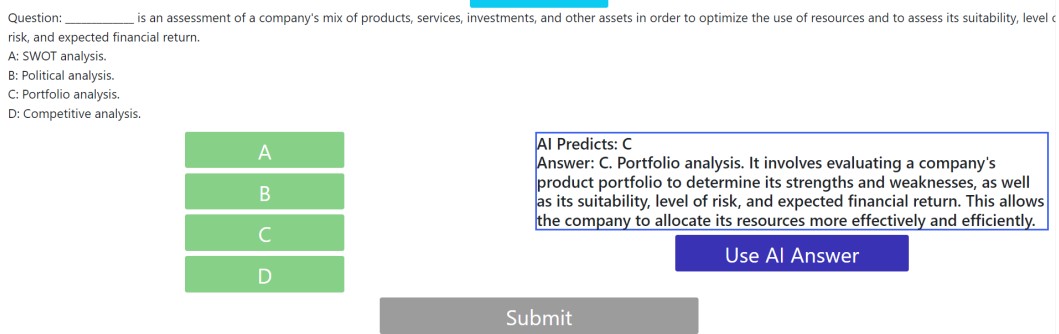

Figure 22: Prediction Interface for MMLU study

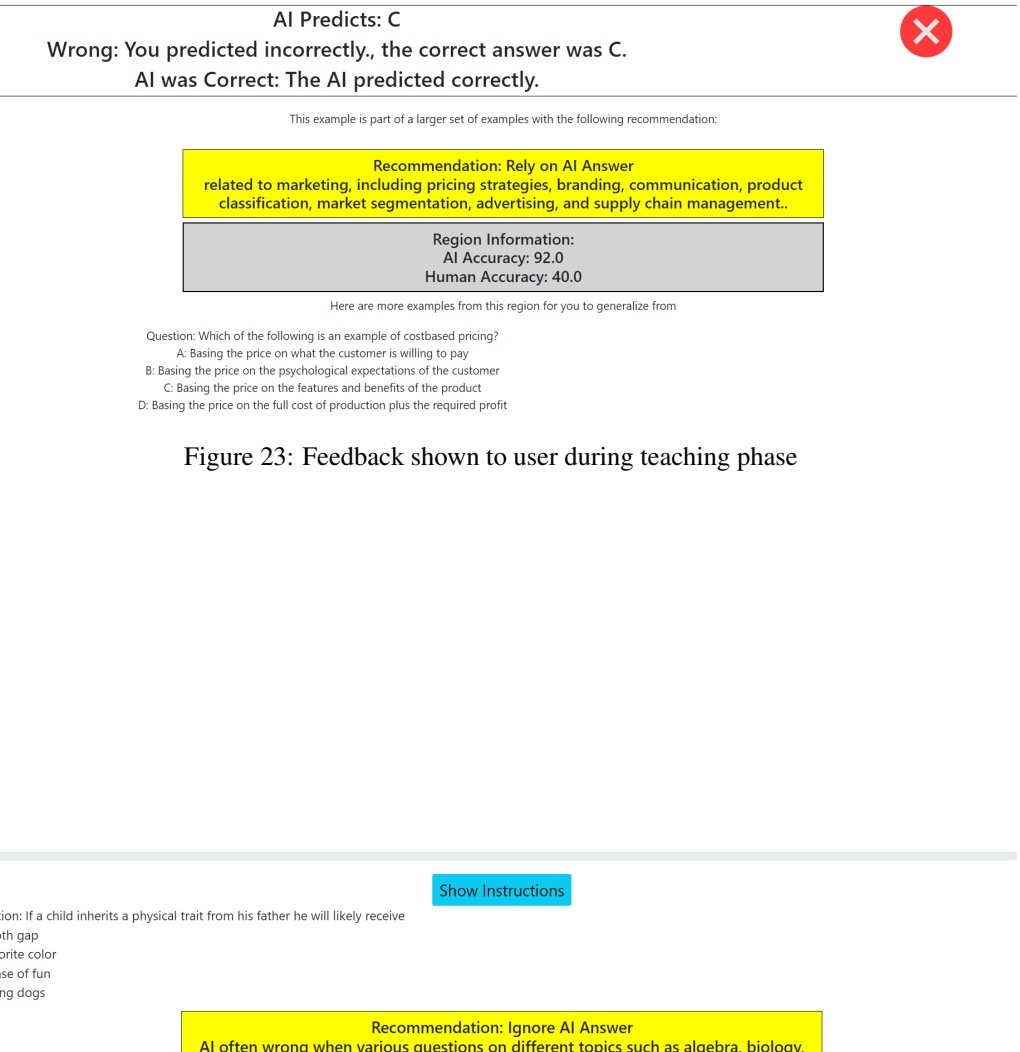

**AI Predicts: C**
Wrong: You predicted incorrectly., the correct answer was C.
AI was Correct: The AI predicted correctly.

This example is part of a larger set of examples with the following recommendation:

**Recommendation: Rely on AI Answer**
related to marketing, including pricing strategies, branding, communication, product classification, market segmentation, advertising, and supply chain management..

**Region Information:**
AI Accuracy: 92.0
Human Accuracy: 40.0

Here are more examples from this region for you to generalize from

Question: Which of the following is an example of costbased pricing?
A: Basing the price on what the customer is willing to pay
B: Basing the price on the psychological expectations of the customer
C: Basing the price on the features and benefits of the product
D: Basing the price on the full cost of production plus the required profit

Figure 23: Feedback shown to user during teaching phase

Show Instructions

Question: If a child inherits a physical trait from his father he will likely receive
A: tooth gap
B: favorite color
C: sense of fun
D: liking dogs

**Recommendation: Ignore AI Answer**
AI often wrong when various questions on different topics such as algebra, biology, public relations, and statistics with multiple options to choose from.

A

B

C

D

AI Predicts: A
The answer is A. tooth gap because physical traits, such as the gaps between teeth, are primarily determined by genetics and can be passed down from parents to children.

Use AI Answer

Submit

Figure 24: Prediction Interface for MMLU study with AI-integration recommendations.

