# OpenReview forum: "Effective Human-AI Teams via Learned Natural Language Rules and Onboarding"
_NeurIPS.cc/2023/Conference — NeurIPS 2023 spotlight_

### Official Review · Reviewer_YAuF · 2023-07-06

**Soundness:** 2 fair
**Presentation:** 3 good
**Contribution:** 4 excellent
**Rating:** 6
**Confidence:** 4

**Summary:**

The authors propose an HAI system called IntegrAI that should help people make better decisions about when to defer to AI predictions, make predictions themselves, or combine their decisions.
As part of their system, the authors perform semantic clustering using LLMs to find areas of disagreement between AI and humans and distill those clusters into simple rules that can be surfaced as insights to users.

The IntegrAI-describe algorithm can be summarized as:
1. Create clusters in joint embedding space
2. Get candidate descriptions of each cluster and embed them into same space
3. Search for counterexamples
4. Update clusters and repeat

The key claims are:
1. IntegrAI learns optimal integration decisions
2. Rules generated by the oracle (GPT3.5 in this case) are easily understandable
3. Onboarding calibrates human expectations about AI performance
4. Dashboard insights help humans choose which action they should take thereby improving teaming performance.

**Strengths:**

- the IntegrAI framework is novel and very interesting
- the IntegrAI-describe algorithm is a very clever way to leverage LLMs for semantic clustering

**Weaknesses:**

1. Several of the main claims (see list in summary above) are unsubstantiated by empirical results. Specifically:
- claim 2 (rules are easily understandable) is not extensively tested and the results comparing it against SEAL seem inconclusive.
- experimental results provide only weak evidence for claim 3 and claim 4, and it remains unclear whether a combination of Onboarding + Rec is actually useful (e.g. if looking only at effect size means, then Onboarding alone has both higher performance and lower time than Onboarding+Rec)

2. Additionally, I think there may be some issues in the significance testing for the results in Table 3. Given the large reported standard errors, it seems unlikely that the p-values would be as small as reported.

Minor suggestions:
- small typo in figure 1: no panel "d)"
- small type on line 57: should be "compared" instead of "compare"

**Questions:**

See sections above.

**Limitations:**

The proposed framework is a 2nd-order framework in the sense that:
- 0th order is if human or AI individually make decision
- 1st order is if human sees AI decision and can change mind
 - 2nd order is another AI sits on top of this process and tells human whether they should change their mind.
For this type of framework to be useful, there needs to be a complex setting where humans struggle to model their own uncertainty/limitations as well as those of the AI. I think the traffic light setting may not be adequately complex to really let this framework shine and show off its capabilities. I think it would really strengthen the paper and provide evidence for the key claims if the authors could evaluate on a more complex setting (e.g. multi-class, greater ambiguity, etc.)

Also, the proposed approach requires collecting paired data. This is a potential limitation if a lot of expensive human data is required. It would be great to see a discussion on how much paired data is required and whether this is a major limitation. An additional limitation of the paired data collection procedure the authors use in their particular case study is that integration can't be detected (i.e. currently authors only detect whether the human used the AI answer or didn't).

---

> ### Author Rebuttal · Authors · 2023-08-10
>
> Thank you so much for your detailed review, we appreciate your time and all the important comments.
>
>
> **Error Measurements Typo**: Before we delve into each point in your review, we want to clarify a typo in our paper. We report standard deviations instead of standard error  for all results when we write $ XX.X \pm X.X$. Just to clarify in detail what we mean: given a random variable X (e.g. performance of human on user study), given samples $x_1,\cdots,x_n$ we can compute the mean $\mu_n$ (np.mean) and the standard deviation $\sigma_n$ (np.std as used in our code in the supplement) measures the deviation around the mean. On the other hand, the standard error is computed as standard deviation divided by the square root of sample size: $e_n=\sigma_n / \sqrt{n}$ measures the precision of our mean measurement $\mu_n$ to the true mean $\mu$.
>
> ## Weaknesses:
>
> 1. We agree that we should make a better job to make clear that claims 1-4 are hypotheses that we as authors made before developing the experiments and are then either refuted or backed by the experiments, the introduction states these hypotheses which then section 6-7 help rebute or back.
>
> *“claim 2 (rules are easily understandable) is not extensively tested and the results comparing it against SEAL seem inconclusive.”*:
>
>  It is true that we did not perform direct quantitative analysis about the rules being interpretable themselves, what we did test instead in the user study is whether the rules presented through onboarding have an effect in improving the human accuracy. Moreover, we have in the appendix show examples of these rules for BDD and for MMLU in Tables 7 and 8 respectively so one can qualitatively evaluate whether they are intepretable. Finally, the results in aim 3 (sec 6) evaluating our method compared to SEAL do show a noticeable effect: with respect to METEOR score, our method achieves 0.32 $\pm$ 0.04  (standard error: $0.04 = 0.30 / \sqrt{50} $ where 50 is the total amount of objects tested for Aim 3) while SEAL achieves 0.10 $\pm$ 0.02.
>
> *‘experimental results provide only weak evidence for claim 3 and claim 4“:*
>
>  We believe our user study provides strong evidence for claim 3 in the task studied as we have a statistically significant effect for onboarding compared to baselines. For claim 4, as the results show, that claim is refuted as clearly recommendations increase the time to make a prediction and do not improve performance beyond onboarding. We will make in future iterations of this paper that the claims in the intro are hypotheses that we test and then include the result of our tests from the beginning.
>
>
>
> 2.  We hope that the note at the beginning of the rebuttal clarifies the issue with standard errors (we reported standard deviation). There are no issues at all with the significance testing performed. Table 3 with standard errors instead of standard deviation is as follows (simply divide the standard deviation by square root of 50, as 50 is the sample size)
>
>
> | Metric           | AI only        | Human          | Human-AI       | Onboard(ours)+Rec | Onboard(ours)  | Onboard(baseline) | Rec              |
> |------------------|----------------|----------------|----------------|-------------------|----------------|-------------------|------------------|
> | Accuracy (\%)    | $79.2 \pm 1.3$ | $78.5 \pm 1.8$| $77.2 \pm 1.5$| $81.7 \pm 1.2$    | $82.9 \pm 1.2$ | $79.9 \pm 1.4$    | $81.4 \pm 1.7$ |
>
>
>
>
> ## Limitations:
>
> *“The proposed framework is a 2nd-order framework in the sense that:”*
>
> Practically our interface is a 1st order framework in your terminology and not a 2nd order framework since the human sees the AI decision and then makes the final decision, the AI does not tell the human to change their opinion (interface in Figure 3). However, it can be considered as a pseudo-2nd order framework since during onboarding we use the data of the human relying on the AI and try to correct the human’s perception about the AI.
>
> *“I think it would really strengthen the paper and provide evidence for the key claims if the authors could evaluate on a more complex setting (e.g. multi-class, greater ambiguity, etc.)”*
>
> -> Our user study task on traffic light detection went through multiple iterations to make sure the task is interesting which was accomplished with varying increasing levels of blur to the images to accomplish “greater ambiguity” as you mention.
>
>
> *“approach requires collecting paired data.”*
>
> This is a major limitation of our work in that to work optimally we need human data of interacting with the system. The approach can be made to function without paired data but performance would decrease. We go around the limitation of limited paired data by building predictors of the human prior from the limited data we have so that we can essentially increase the sample size but might introduce additional bias. We will add this as a limitation in our paper in the next iterations. Finally, while it is true that integration cannot be detected which we fully acknowledge in lines 148-151, human prediction with and without AI can reveal some information of integration.

---

> > ### Comment · Reviewer_YAuF · 2023-08-10
> > **Clarification on rebuttal**
> >
> > Thank you for the detailed response!
> >
> > ## Claims
> > I'd like to make sure I understand the key claims you are making. For example, on lines 34-36 you write: "We further propose to surface the AI-integration decisions found by IntegrAI as recommendations to the human within an AI dashboard used after onboarding. This AI dashboard helps the human evaluate which action they should take, thereby leading to effective AI adoption for enhanced decision-making." and on lines 230-231 (in a section titled "Onboarding and Recommendations to Promote Rules") you write: "We accomplish this through an onboarding process followed by test-time recommendations, as described next." and at the end of section 5 in line 262 you also write "These recommendations are shown as an aid to the human to reduce their decision-making time." all of which I summarized as claim 4. In that case, your full proposed method would be Onboarding (ours) + rec.
> >
> > My understanding based on your response is that you are changing all of these claims about dashboard/rec now to instead be hypotheses, which are then refuted in the last few sentences of your analysis.
> >
> > The updated claims are then that the dashboard is actually not helpful (i.e. rec increases time but not performance) and that your proposed method is just Onboarding (ours)? Would you say your contributions are then 1) a novel method for learning rules (i.e. the content); and 2) a novel onboarding method (i.e. the delivery method)? Is rec just a baseline then that acts as an ablation of the delivery method since it involves showing rules to people but not onboarding them? Totally fine for rec to be a negative result by the way, I strongly believe we shouldn't punish authors for including negative results since these can also be beneficial for the community.
> >
> > If that's the case, then:
> > - comparing "Onboarding (ours)" vs "Onboarding (baseline)" would test whether the rules you come up with are actually helpful.
> > - comparing "rec" vs "Human-AI" would also test whether the rules you come up with are actually helpful.
> > - comparing "Onboarding (ours)" vs "rec" would test whether the onboarding procedure you came up with is actually helpful.
> > - comparing "Onboarding (baselines)" vs "rec" would also test whether the onboarding procedure you came up with is actually helpful.
> > - etc.
> >
> > I think these comparisons (though many of them won't be statistically significant) reveal the following insights that disentangle what each component contributes:
> > - The rules you come up with lead to increased performance
> > - The onboarding procedure leads to decreased time (worth noting that this is at the cost of increased training time, so this only becomes worthwhile for tasks that need to be repeated many times)
> >
> > ## Statistical testing
> > Thank you for the correction, the test results make a lot more sense if those were standard deviations in the table though I think there may still be typos or some other problems. For example, shouldn't the p-value for Onboard (ours) vs Human-AI be 0.004 rather than 0.0004.
> > Also weren't 6 tests conducted in total rather than 5 (affecting the Bonferonni correction)? Finally, if you are relying on statistical significance testing to substantiate your claim, shouldn't you also test other claims you make like that "Onboarding (ours)" outperforms "Onboarding (baseline)" and "AI only"?

---

> > > ### Author Response · Authors · 2023-08-13
> > > **Response to Reviewer**
> > >
> > > We thank the reviewer for such detailed reading of our paper and our rebuttal. We are mostly in agreement with your reading of the paper, we highlight some important details here.
> > >
> > > To forward this comment, we plan to release all code and raw data of our study allowing for easy replication of all results and tables.
> > >
> > > Thank you for following up on Table 3, indeed we realized that the p-values for Onboard(ours)+Rec and Onboard(ours) had an issue in Table 3. The processing code of the raw data for the onboarding (ours) condition was run twice duplicating values. To explicitly point out the error, let X be the dataset of values for Onboard(ours) and Y be the dataset of values for  Onboard(ours)+Rec used to report Table3 in the paper. Let X' and Y' be the true (correct) dataset values for Onboard(ours)  and Onboard(ours)+Rec. The incorrect data relate to the correct data with X=(X',X') (twice duplicated) and Y=(Y',X') (duplicated with the Onboard (ours) values).
> > >
> > > The following is the corrected form of Table 3 (avg and standard deviations reported), all scores are in [0,1] as fractions.
> > >
> > >
> > > | Metric           |      Human          | Human-AI       | Onboard(ours)+Rec | Onboard(ours)  | Onboard(baseline) | Rec | AI only|
> > > |:--------------------------------------|:---------------|:---------------|:---------------------|:-------------------|:--------------------------|:-------------------|:-------------------|
> > > | Accuracy (mean, std dev)              | (0.785, 0.127) | (0.772, 0.106) | (0.804, 0.089)       | (0.829, 0.086)     | (0.799, 0.096)            | (0.815, 0.121)     | (0.792, 0.092)     |
> > > | Test with Human+AI (p-value, t-value) | (0.455, 0.752) | (0.455, 0.752) | (0.1034, 1.64343)    | (0.00394, 2.95053) | (0.20123, 1.28688)        | (0.06136, 1.89218) | (0.14883, 1.44748) |
> > > | AI reliance   (mean, std dev)                         | N/A            | (0.165, 0.227) | (0.672, 0.166)       | (0.256, 0.237)     | (0.238, 0.287)            | (0.211, 0.218)     |                    |
> > > | Time per example    (mean, std dev)                   | (5.408, 2.142) | (7.78, 3.834)  | (7.622, 2.679)       | (5.936, 2.076)     | (6.841, 3.644)            | (8.717, 5.0)       |
> > >
> > > In the corrected table, the accuracy (and std dev) for  Onboard(ours) is the same as in the reported table since duplicating data won't have an effect, the p-value is also only shifted by one digit to the left (0.004). The accuracy of  Onboard(ours)+Rec was 81.7 which we can verify is equal to = (80.4 (correct value for Onboard(ours)+Rec  ) + 82.9)/2, which confirms our error. This error in the analysis was completely unintentional since it wouldn't have changed the significance of our results.
> > >
> > > From your rebuttal and suggested tests, we run them and perform p-value adjustments for multiple testing using the Benjamini Hochberg procedure:
> > >
> > > |    | Condition 1           | Condition 2           |    p-value |   adjusted_p-value |
> > > |---:|:------------------|:------------------|-----------:|-------------------:|
> > > |  0 | Rec               | Human-AI          | 0.0613596  |          0.224985  |
> > > |  1 | AI only           | Human-AI          | 0.148829   |          0.265314  |
> > > |  2 | Onboard(baseline) | Human-AI          | 0.201229   |          0.27669   |
> > > |  3 | Onboard(ours)+Rec | Human-AI          | 0.103404   |          0.262999  |
> > > |  4 | Onboard(ours)     | Human-AI          | 0.00394331 |          0.0433764 |
> > > |  5 | Human             | Human-AI          | 0.573388   |          0.573388  |
> > > |  6 | Onboard(ours)     | Onboard(baseline) | 0.119545   |          0.262999  |
> > > |  7 | Rec               | Onboard(ours)     | 0.51309    |          0.564399  |
> > > |  8 | Rec               | Onboard(baseline) | 0.495399   |          0.564399  |
> > > |  9 | AI only           | Onboard(ours)     | 0.0108641  |          0.0597526 |
> > > | 10 | Onboard(ours)+Rec | Onboard(ours)     | 0.168836   |          0.265314  |
> > >
> > > This performs all pairwise tests between the 3 main conditions (Rec, Baseline Onboarding, and our Onboarding (w/out rec), and tests whether the addition of recommendation to onboarding has an effect. We find that the only significant effect is that of  Onboard(ours)  improving performance over Human-AI without onboarding, all other effects are not significant at level $\alpha=0.05$ (family-wise error rate). Our main claim in the paper about significance in the paper is that our onboarding procedure (without rec) outperforms the Human-AI team, compared to AI-only is close to being significant (0.06) but misses the 0.05 threshold after multiple testing corrections (11 tests).
> > >
> > > Our goal in the paper was to mainly investigate whether our procedure would improve the performance of the human-AI team, and there is only an implicit comparison with the baseline which does not improve performance (we had never performed tests beyond those reported in the paper before this rebuttal as we had internal pre-registration).

---

> > > > ### Comment · Reviewer_YAuF · 2023-08-20
> > > >
> > > > Most of my concerns have now been addressed.
> > > > In particular, thank you for taking my concern about the p-values seriously and thoroughly revisiting your data analysis pipeline.
> > > > Just to make sure everything is updated after that error was corrected:
> > > > 1. did the error you described also affect "AI reliance" and "Time per example" numbers?
> > > > 2. did the error affect both the "Onboarding (ours)" and "Onboarding (ours) + rec" columns? I see you made changes in both of them in the new table (e.g. updated p-value from 0.0004 to 0.004 for "Onboarding (ours)").

---

> > > > > ### Author Response · Authors · 2023-08-20
> > > > > **Response**
> > > > >
> > > > > 1- (Edited) no the error only affected the accuracy column, only the predictions were duplicated, but time and AI reliance decisions were and are correct.
> > > > >
> > > > >
> > > > > 2- the error affected both the "Onboarding (ours)" and "Onboarding (ours) + rec" columns as they come from the same participants (it is one unique condition where each participant does both sub-conditions as mentioned in the paper) and the processing code was the same for both.
> > > > >
> > > > > We thank the reviewer for their diligence which improves the quality of our work, as noted, we will make all our raw data and analysis code publicly available for replication.

---

> > > > > > ### Comment · Reviewer_YAuF · 2023-08-20
> > > > > >
> > > > > > Could you please share the full updated table? I believe in that case the standard deviations should have changed for every entry in both columns and the means should have changed for every entry in the "Onboarding (ours) + rec" column.

---

> > > > > > > ### Author Response · Authors · 2023-08-20
> > > > > > > **Response**
> > > > > > >
> > > > > > > Apologies, the error only affected the accuracy column, only the predictions were duplicated, but time and AI reliance decisions were and are correct.
> > > > > > >
> > > > > > > Here are all the discrepancies you should note and (why they occurred) between the updated table (in the previous rebuttal) and in the submitted version.
> > > > > > >
> > > > > > > - Accuracy for "Onboard(ours)+Rec" "Onboard(ours)", Reason: noted above
> > > > > > > - t-test value for "AI only" column:  the AI only column combines the AI predictions across all instances that participants see, so the error from Onboard (ours) slightly alters the value of the t-test - a very minor change
> > > > > > > - minor change in "Rec" column results: removal of data of one participant that failed 2/2 attention checks (due to random test cases this participant saw 2 instead of 3 attention checks, so the filter for the attention checks passed the participant incorrectly, participants are required to have >50 attention check accuracy to be included)

---

> > > > > > > > ### Comment · Reviewer_YAuF · 2023-08-20
> > > > > > > >
> > > > > > > > I'm a bit confused by this. The table above has the same accuracy (both mean and std) for "Onboard (ours)" as the submitted paper.
> > > > > > > > Only the t-test results have changed in the "Onboard (ours)" column and the change was an order of magnitude 0.0004->0.004.

---

> > > > > > > > > ### Author Response · Authors · 2023-08-20
> > > > > > > > > **Further Explanation**
> > > > > > > > >
> > > > > > > > > As mentioned in the response https://openreview.net/forum?id=V2yFumwo5B&noteId=lioEoRq57a  (one with the table),
> > > > > > > > > the data used to calculate the accuracies for "Onboard (ours)" was twice-duplicated, each entry was duplicated twice, so instead of an array x of 50 accuracies (for each participant), we have an array x' of 100 accuracies where x'=(x,x).
> > > > > > > > >
> > > > > > > > > The mean of x and the mean of x' should be the same, $mean(x') =  2* \sum_{i}^{50} x_i / (2 * 50) =  \sum_{i}^{50} x_i /  50 = mean(x)$, similarly for standard deviation: $ std(x') = \sqrt{2*\sum_{i}^{50} (x_i - mean(x') ) / (2*50)}  = \sqrt{\sum_{i}^{50} (x_i - mean(x) ) / (50)} = std(x)$
> > > > > > > > >
> > > > > > > > > Let us recall the t statistic used in our p-value computation (two sample t-test) between two sets of samples $x_1$ and $x_2$:
> > > > > > > > >
> > > > > > > > > $$ t = \frac{{\bar{x}_1 - \bar{x}_2}}{{\sqrt{\frac{{s_1^2}}{{n_1}} + \frac{{s_2^2}}{{n_2}}}}}$$
> > > > > > > > > - \(t\) is the test statistic.
> > > > > > > > > - \(\bar{x}_1\) and \(\bar{x}_2\) are the sample means of the two groups.
> > > > > > > > > - \(s_1^2\) and \(s_2^2\) are the sample variances of the two groups.
> > > > > > > > > - \(n_1\) and \(n_2\) are the sample sizes of the two groups.
> > > > > > > > >
> > > > > > > > >
> > > > > > > > > We can see that the p-value relies on three things: mean, standard deviation and sample size. Since our sample size was doubled, only the p-value  and the test statistic t will change.

---

> > > > > > > > > > ### Comment · Reviewer_YAuF · 2023-08-20
> > > > > > > > > >
> > > > > > > > > > Ahh I see you're using population standard deviation rather than sample, that explains it.
> > > > > > > > > >
> > > > > > > > > > My concerns about key claims and results reporting have been resolved and I'm updating my score accordingly.

---

### Official Review · Reviewer_Ea7z · 2023-07-07

**Soundness:** 3 good
**Presentation:** 2 fair
**Contribution:** 3 good
**Rating:** 5
**Confidence:** 3

**Summary:**

This paper discusses some research problems a very interesting scenario where human and AI need to collaborate to achieve a certain goal. The authors propose to learn rules grounded in data regions and described in natural language, which illustrates how the human should collaborate with the AI agent.

The paper also proposes a novel algorithm, region discovery algorithm, which uses an iterative procedure where a large LM describes the region while distinguishing it from the rest of the date. Experiments show positive results when following this protocol.

**Strengths:**

1. It tries to solve research problems in a very interesting domain.
2. The user study is extensive and its setting is convincing.

**Weaknesses:**

1. The presentation needs huge improvement. e.g., some figures are not clear and text is vague.
2. The study was conducted with 25 human participants, which might not be representative human users.


**Questions:**

Have you considered about the "cost" during human AI collaborations? e.g., time. If fully automated pipeline can work well with less cost, that might be our preferred one.

**Limitations:**

The relatively small scale human evaluation.

---

> ### Author Rebuttal · Authors · 2023-08-10
>
> Thank you so much for your detailed review, we appreciate your time and all the important comments. Please read below for our response.
>
> ## Weaknesses:
>
> 1. We will improve the presentation and clarify of the paper in our next iteration. In particular, we will clarify the methods in section 4 better and figure 1 and others.
>
> 2. *"The study was conducted with 25 human participants, which might not be representative human users."*
>
> Please note that the final user study was conducted with *150* human participants, with 50 humans in each condition. The initial 25 human participants were used for data collection to build the approach, then an additional 150 participants were recruited. This number of participants per condition is comparable to prior work in the literature and was sufficient to establish the statistical significance of our approach.
>
> ## Questions:
>
> Thank you for an insightful question. We did indeed consider a form of cost of the human-ai interaction, in particular time to make a prediction as you suggest. We found in Table 3 (last row), that Humans without the AI spend 5.4s per example, without onboarding they spend 7.78s per example, however with onboarding this reduces to 5.9s which is a significant decrease (two sample t-test, p=0.0014, t=3.2872).
>
> ## Limitations:
>
> Please note again that the user study evaluation was conducted with 150 participants (50*3) as mentioned in lines 360-362 of section 7 “Experimental Conditions”.
>
> If our rebuttal changed your view of the paper given the corrected sample size of participants for the studies, please consider adjusting your score accordingly.

---

> > ### Author Response · Authors · 2023-08-17
> > **Author Rebuttal**
> >
> > Dear Reviewer, I hope you get the chance to read our rebuttal that tries to address your concerns, if it does, please consider adjusting your review in response.

---

> > > ### Comment · Reviewer_Ea7z · 2023-08-19
> > > **Thanks for your response!**
> > >
> > > Thanks for the author's response, and I think with better presentation the paper would be more readable and can be better interpreted by the readers from a variety of backgrounds.
> > >
> > > I would like to keep my original rating.

---

### Official Review · Reviewer_oGDF · 2023-07-08

**Soundness:** 3 good
**Presentation:** 3 good
**Contribution:** 3 good
**Rating:** 8
**Confidence:** 4

**Summary:**

This paper presents an approach to allow effective human-AI teaming. The approach describes a process of semantic discovery of regions in an embedding space following by generating natural language descriptions of the regions and then learning refinement of the regions using counterexamples. Once regions and region descriptions are generation, human onboarding is performed alongwith proposals for whether to rely on AI for decision making or not with an explanation. Analysis shows that human-AI team performs better than just AI or humans. User study shows very interesting results (role of onboarding, role of providing recommendations, impact of time per task with/wo AI teaming and recommendations).

**Strengths:**

- This paper presents an interesting and practical approach for developing a human AI collaboration system with a focus on both machine learning aspects (learning AI integration functions) and the HCI aspects (onboarding + natural language descriptions) of how human-AI teams can really work in balance (staying away from over-reliance on AI and under-reliance on AI).
- The use of chatGPT as an oracle is also a great workaround for lack of ground truth data
- Evaluation shows that the learnt AI integration function leads to lowest loss at test time and also minimum number of region proposals (which may not overwhelm the user)
- Groundtruth cluster analysis of regions on the BDD dataset also shows high correlation compared to other region discovery approaches.
- the contrastive approach for region description is also very intuitive and gives good results.

**Weaknesses:**

- There's a lot covered in the paper in the limited space and the focus seems to be on region discovery and region description algorithms as well as the higher level human-AI teaming and overall system level evaluation study. It seems like one could do more extensive analysis with additional experiments with more datasets on each of the sections.


**Questions:**

- There are other approaches for region discovery such as the deep aligned clustering algorithm, adaptive decision boundary algorithms etc (from the fields of open class learning, semi-supervised representation learning, novel intent discovery, etc) What do the authors think about these other approaches for region discovery.

**Limitations:**

- This paper covers a lot of ground and the authors have also captured the current limitation well in the Limitations section of the paper. However, given all the limitations, the paper presents some very promising ideas and directions of further research on human AI collaboration.

---

> ### Author Rebuttal · Authors · 2023-08-10
>
> Thank you so much for your review, we appreciate your time to review our paper and the important considerations you raise.
>
> Weaknesses:
>
> We agree that we can add more experiments on more datasets. We do plan to add more ablation experiments and on more datasets in the appendix (on ImageNet16H, CIFAR-10H and a synthetic gaussian setup) notably.
>
> Questions:
>
> Thank you so much for pointing out two related works that can be applied for region discovery [1,2] that we will add a comparison to and cite in future iterations of our work.
> In [2] the authors present a novel method for intent classification that leverages prior knowledge of previously known intents and tries to discover new intents (analogy between intent and regions in our work). In our work, there are no known previous regions so we cannot leverage the prior data in that fashion as the authors use to fine-tune the representation. As in [2], we do not know the number of regions a priori, and their approach to filter clusters can be useful. We find that the deep aligned clustering algorithm to be very interesting to apply a baseline in our tasks. One caveat is that it would require changing the cross-modal embeddings (CLIP) we use and it has been shown by DOMINO [17] that they outperform other uni-modal representations such as BERT embeddings for region discovery. Similarly, we also think that the Adaptive Decision Boundary Learning algorithm in [1] can also be applied to discover regions and might work well on the setting. We look forward to applying both algorithms as baselines in future iterations of our work.
>
>
>
> [1] Zhang, Hanlei, Hua Xu, and Ting-En Lin. "Deep open intent classification with adaptive decision boundary." Proceedings of the AAAI Conference on Artificial Intelligence. Vol. 35. No. 16. 2021.
> [2]  Zhang, Hanlei, et al. "Discovering new intents with deep aligned clustering." Proceedings of the AAAI Conference on Artificial Intelligence. Vol. 35. No. 16. 2021.

---

### Official Review · Reviewer_doL5 · 2023-07-09

**Soundness:** 3 good
**Presentation:** 3 good
**Contribution:** 3 good
**Rating:** 6
**Confidence:** 5

**Summary:**

The authors attempt to improve human-AI collaboration through a system that (1,2) identifies similar regions in a dataset (both in terms of the task similarity and in terms of human behavior—usage of AI advice—on the task) and optimizes for how humans should ideally use the AI in the given region, (3) generates contrastive descriptions for the region, and (4) provides recommendations to users based on 2. They synthetically evaluate each component of their system, and empirically test their system with human users across two tasks.

**Strengths:**

1. Problem setup and optimization solution are straightforward and make sense.
2. Empirical analysis using humans to assess their system (with multiple datasets).
3. Generally well-written. See “Weaknesses” for a few suggestions I have to clarify some parts (section 4).


**Weaknesses:**

1. Only testing on one task
2. The experiment design has an obvious confounder — the raters are only offered a binary choice, while the AI
3. The setup did not allow distinguishing between ignoring the AI and collaborating with the AI. This should be possible with an adjusted setup — for example, have a 2 stage process for data collection where the person makes an initial decision and subsequently is given the chance to use AI (in section 7, it sounds like you did this, but this was not clear earlier in the paper). Additionally, rather than having people submit a discrete label, you can ask them to submit a continuous value (e.g., something like a confidence). Note that this could be done just for the dataset collection process to approximate $R^*$, and you could still use your original setup for the empirical evaluation. (Side note: if you had recorded a “confidence” for the dataset, optimizing for the regions may be easier as this is a differentiable problem now since the $r$ would be continuous-valued.)
4. It’s actually unclear how well this setup can distinguish between 0 and  1 — using the AI vs not using the AI. Since everyone sees the AI’s output, they may be biased based on that while still making their own decision. There are a number of studies that have established the biasing effect this can have.
5. The overall method in 4 (and 4.1, 4.2) is relatively straightforward, but it is a bit difficult to parse initially (some key points are buried in the details). I would recommend a concise description at the start of each section to describe what will be done in that section. (e.g., at the start of 4.1, saying something like “We generate an embedding for each datapoint by concatenating a task embedding with the human’s decision to use AI. We then define an objective function to jointly discover data regions and optimal usage of AI; it seeks to maximize task instance similarity while enforcing similar (optimal) human behavior (usage of the AI) on that task”). In my opinion, this high level overview will make the sections easier to read.
6. In the appendix, it seems the empirical analysis on the second dataset results in not significant benefits to using the proposed system. This is OK, but I would recommend adding discussion about this in the main paper. It is important to understand when your system may and may not work, and where improvements may be needed. For example, it is possible the disparity in human and AI accuracy is large enough that simply trusting the AI is a better strategy than trying to assess when to trust the AI more.

**Questions:**

1. How important is the embedding choice for the region discovery algorithm?
2. There seem to be a lot of hyperparameters here. How important are the choices here? For example, the upper and lower bound on fraction of points in a region seem to be an important consideration — were there any experiments to assess what is reasonable for (a) human users and (b) for the generated region description? How many regions is reasonable for onboarding?
3. For the region parametrization, are the CLIP embeddings normalized?
4. For region description, (1) the initialization and (2) the embedding could matter a lot. Did you do any analysis of the variation in the final description with respect to these?
5. What do you use to generate the initial text descriptions for each image? In 4.2 you reference a few possible methods, but don’t indicate which you use.
6. In Table 1, I am unclear on what the numbers represent. How is the error calculated?
7. For the empirical studies, is the task order randomized? Also, for the people who did “predicts alone” and “predicts with help of AI”, were the tasks where they predicted alone randomized person to person?

**Limitations:**

As the authors recognize, it would always be nice to have more empirical studies (e.g., with more users, more datasets, etc.). The authors also recognize the potential negative outcomes of this work (incorrectly identified regions and/or recommendations could incorrectly bias users to under/over-trust the AI).

---

> ### Author Rebuttal · Authors · 2023-08-10
>
> Thank you so much for your detailed review, we appreciate your time and all the important comments.
>
> ## Weaknesses:
>
> 1. It is true that we test our onboarding method on one user study task in the main body of the paper.  However, the purpose of the user study is to test the behavior of various humans (150 overall) when they undergo onboarding. In the setting studied, we showed that onboarding can have a positive effect on performance, we believe the conclusion would remain the same if the task was different (e.g., classify species of animals) but with the same explanations, and relative human and AI performance individually. We do agree additional tasks would be interesting to see how the effect of onboarding changes with different task parameters, this is left to future work. Most importantly note that we do evaluate the components of our method on four different datasets and different tasks within section 6.
> 2. It seems the sentence of this point was cut in the middle, we are more than happy to reply to the full point if re-written during the discussion period.
> 3. With this setup, we cannot (as you correctly point out) always identify which integration decision did the human take, we can only do it when the press the "use AI answer" button. This is something we clearly acknowledge in lines 146-150, this means we can only learn a proxy version of the human integrator. Let us clarify that our data collection in the user study consists of two parts: 1) human predicts without AI on a set of examples and 2) human predicts with AI using the same interface as the evaluation interface. While it is true that we can devise other forms of data collection schemes, notably collecting human confidence which is something we considered but did not pursue as we believed it would be too subjective and create additional biases, we were mostly interested in figuring out how to learn from feedback data collected from the same interface participants will predict on at test time.
> 4. It is correct that it is hard to distinguish which integration decision the human took, the only sound way to do this is to ask the human explicitly after they predict which would be an interesting study design but not as relevant to learning accurate regions which do not require exact knowledge of the human prior.
> 5. Thank you so much for pointing out how we can better improve the presentation of section 4, we will improve the presentation in future iterations.
> 6. The MMLU results had only been completed after the full paper deadline due to budget constraints, so in the main paper, we only mentioned performing the study but did not elude to the results (whether positive or negative we had planned to include them in the main paper if they had been done in time). Our hypothesis for why the MMLU onboarding results are inconclusive is that the ChatGPT explanations are very informative about the model’s uncertainty, in fact, ChatGPT frequently responds with an explanation stating “I don’t know” or “there is not enough information”. To test a weak version of this hypothesis, we ran a follow-up study conducted after the Appendix deadline with Human-AI performance without ChatGPT explanation but only the ChatGPT answer. We found the Human-AI performance drops to 70.4% (no explanation) compared to 75.1% (with explanation) (from Appendix table 9), a two-sample t-test finds p =0.045 and t=2.03.  Thus our hypothesis is that onboarding is most effective when the AI explanation do not inform the user when and when not to rely on the AI.
>
> ## Questions:
>
> 1. The choice of the embedding space for region discovery algorithms has been established by prior work DOMINO [17] and has a significant effect on the ability to recover regions of high error. The authors in [17] found that cross-modal embeddings work best compared to random initialization, uni-modal embeddings, and activations of the model itself. We follow the recommendations of prior literature and use the CLIP cross-modal embeddings.
> 2. There are four main hyperparameters for the region-finding algorithm: $\alpha$ (0.5 used) controls consistency of the region, $\beta_l$ (0.01 used) and $\beta_u$ (0.1) control the minimum and maximum size of the region and $\delta$ (1 used)  minimum gain of the region. The two parameters that have the most impact are $\alpha$ and $\beta_u$ which we experimented with quite a bit. Setting $\beta_u$ to 1 (unconstrained) produces few regions of large size and a large number of tiny regions so that the choice of 10% size leads to most often a dozen good-sized regions. For onboarding, we chose 9 so that they can fit in time within a user study.
> 3. We do not normalize the CLIP embeddings.
> 4. We did not find that the initialization matters a lot for region description since the method allows one to find counterexamples and quickly fix a bad initialization by adding more points.
> 5. We use a different method for each dataset. For BDD, we use the metadata provided with each image to construct a textual description with a template:
>  caption =   caption += scene + " during the " + time_of_day + " with " + weather + " weather, the scene contains " + objects_in_scene
> For MS-COCO we use the captions provided with the dataset.
> For MMLU we use the question itself as the description. For DynaBench we use the review as the textual description.
> 6. Let us first clarify that we report in Table 1 and Table 3 (and in Aim 3) results as mean +- standard deviation. In Table 1, we report the loss (equation 1 with l(.,.) being the 0-1 loss) of the human-AI team when following the recommendations of the AI-integration function learned with the regions.
> 7. In all the user study conditions, for each participant, we select a random set of examples so that each participant per each condition sees different examples. For conditions (1) and (2) described in lines 362-368, we randomize which of the two sub-conditions participants see first for each participant.

---

### Author Rebuttal · Authors · 2023-08-10

Thank you for the reviews of our paper and the insightful suggestions. Please find under each review our rebuttal.

We wanted to iterate the main claims and contributions of our paper:

- We propose a novel region discovery algorithm (sec 4.1) and evaluate it  on four different datasets and show that it can outperform baselines (aim 1 and aim 2 in sec 6)
- We propose a novel region description algorithm (sec 4.2) (automatic cluster labeling), we show quantitatively that it can uncover the ground truth cluster label in an ablation study on MS-COCO (aim 3 in sec 6) and show qualitatively that it produces coherent descriptions (table 7 and table 8 in Appendix)
- We propose an onboarding scheme that presents the regions discovered as “lessons” to the human (step 1,2,3 in section 6), we evaluate this strategy with a full user study with 150 participants on the BDD dataset in section 6 and show that onboarding can significantly improve performance (p=0.002). We additionally propose to show the recommendations of the region as an aid to the human (end of section 6), however, we show that they are not as effective as onboarding.

We are very glad that the reviewers thought that the problem tackled was “in a very interesting domain.” (reviewer Ea7z) and “interesting and practical approach for developing a human AI collaboration system” (reviewer oGDF),  our “framework is novel and very interesting” (reviewer YaUF), the approach to describe regions is clever “IntegrAI-describe algorithm is a very clever way to leverage LLMs for semantic clustering“ (reviewer YaUF) and “use of chatGPT as an oracle is also a great workaround“ (reviewer oGDF) and that the “user study is extensive and its setting is convincing” (reviewer Ea7z).

We also want to point out a typo in the paper on line 376 “$\pm$ standard error”, every result in the paper is reported as mean +- standard deviation (instead of standard error). One can obtain standard error from standard deviation by dividing by the square root of the sample size.

---

### Decision · Program_Chairs · 2023-09-21

**Decision:**

Accept (spotlight)

**Comment:**

The paper presents a new approach to improve human-AI teams, leveraging semantic analysis to identify tasks that humans should rely on AI, and using LLM to generate natural language rules to communicate to humans. The approach is evaluated with human-subject experiments that provided the natural language rules as onboarding methods.

The paper is addressing a timely and important question. The approaches are novel and sound, and the evaluations are overall satisfactory. There are various suggestions from the reviewers, including the presentation and aligning the claims with the results. However, overall, the reviewers all agree that this is a solid work and we recommend acceptance.